# IRGCL: INFORMATION REFINEMENT GRAPH CONTRASTIVE LEARNING

## ABSTRACT

Graph contrastive learning (GCL) has emerged as a leading paradigm in unsupervised graph representation learning (UGRL), yet existing contrastive approaches remain vulnerable to three persistent challenges: noisy features that distort similarity measures, unreliable structures that contain spurious edges, and degree imbalance that biases representation quality. We propose Information-Refinement Graph Contrastive Learning (**IRGCL**), a single-view contrastive learning framework that simultaneously addresses these challenges and effectively generalizes across key graph learning tasks, including node classification, clustering, and link prediction. **IRGCL** integrates three complementary components: (i) structure-consistent feature selection to filter out redundant or noisy attributes; (ii) high-confidence structure learning to refine graph neighborhoods; and (iii) degree-aware focal contrastive learning to balance learning across low- and high-degree nodes. These modules follow a *1–2–N* hierarchy: level-1 node-wise feature refinement, level-2 sample-wise positive-pair refinement, and level-$N$ degree-aware reweighting of node–neighborhood losses. Extensive experiments on diverse benchmarks demonstrate that **IRGCL** consistently outperforms state-of-the-art baselines, and ablation studies confirm the distinct and complementary benefits of each component, highlighting the necessity of jointly addressing feature quality, structural reliability, and degree imbalance. Code is available at
`https://anonymous.4open.science/r/IRGCL-01F8`.

## 1 INTRODUCTION

Graph-structured data is ubiquitous, appearing in domains such as citation networks, e-commerce, and biology. Unsupervised graph representation learning (UGRL) has emerged as a powerful paradigm to extract both structural and semantic patterns without manual labels (Ark et al., 2019; Sun et al., 2019), enabling downstream tasks like node classification (Kipf & Welling, 2017; Veličković et al., 2018), link prediction (Kipf & Welling, 2016; Peng et al., 2020b), and clustering (Wang et al., 2019; Zhang et al., 2019a).

Among UGRL approaches, self-supervised contrastive learning has shown remarkable success. These methods pull semantically or structurally related nodes closer while pushing apart unrelated ones, yielding discriminative

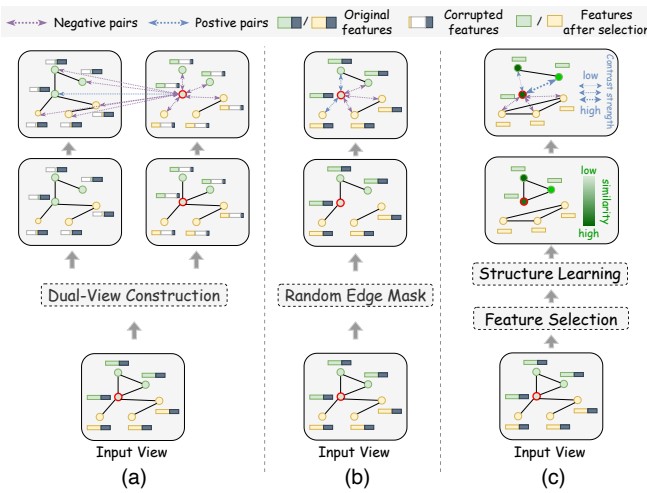

Figure 1: (a) Dual-view contrastive learning (e.g., GRACE); (b) Single-view contrastive learning (e.g., SIGNA); (c) Our information-refinement single-view contrastive learning (**IRGCL**).

and structure-aware embeddings. The main difference lies in how positives and negatives are constructed: dual-view methods (e.g., GraphCL (You et al., 2020), GRACE (Zhu et al., 2020b)) rely on perturbing the graph to form contrasting views, while alternatives avoid explicit augmentation via bootstrapping (e.g., BGRL (Thakoor et al., 2021)) or input–latent contrast (e.g., GMI (Peng et al., 2020a)). More recently, single-view methods (e.g., SIGNA (Sun et al., 2025)) form positives directly from neighborhoods, avoiding fussy augmentations. Figure 1 summarizes these paradigms.

In this work we focus on *single-view, neighborhood-based contrast*, where for each node $u$ the positive set is its one-hop neighbors $P_u = \mathcal{N}(u)$ and negatives are drawn from $V \setminus (P_u \cup \{u\})$. The contrastive objective is then fully determined by pairwise similarities $D_\phi(u, v)$ between node embeddings. Despite strong progress, existing graph contrastive methods remain vulnerable to three persistent challenges that manifest as *failures of this contrastive objective*, rather than mere properties of the underlying graph.

*(i) Feature quality.* Noisy or redundant attributes distort the similarities $D_\phi(u, v)$ that drive the contrastive loss: adjacent nodes may appear spuriously dissimilar, while non-neighbors may look spuriously similar. Thus neighborhood-based contrast is trained on corrupted positive/negative signals. We therefore introduce a *structure-aware low-rank feature selector* that aligns the retained feature subspace with the graph before it is fed into $D_\phi$ (Sec. 3.1).

*(ii) Graph reliability.* By *graph reliability* we mean how well observed edges align with the underlying semantic/label structure (homophily). In reliable graphs, $P_u = \mathcal{N}(u)$ yields clean positives; in unreliable graphs, cross-class edges and missing within-class edges inject false positives/negatives into neighborhood-based contrast. We formalize unreliability via cluster consistency in the embedding space: edges across different high-confidence clusters, or intra-cluster edges with very low silhouette-scaled similarity (below a threshold $\tau$), are treated as *spurious*. This motivates a confidence-guided *structure learning* module that densifies reliable intra-cluster edges and prunes spurious ones (Sec. 3.2).

*(iii) Degree imbalance.* When $P_u = \mathcal{N}(u)$, the size and quality of each node's positive set are degree-dependent. Low-degree nodes have very few, often "easy" positives and receive weak gradients; high-degree nodes aggregate many noisy or mixed neighbors, causing class collision and overfitting around hubs (Hu et al., 2025). Thus the contrastive objective becomes biased across degree ranges. We address this with a *degree-aware focal JSD loss* that adaptively reweights per-node contrastive terms to mitigate degree-induced bias (Sec. 3.3).

These three issues—distorted similarities, unreliable neighborhoods, and degree-biased positives—are tightly coupled through the same single-view contrastive objective. Conceptually, our design follows a hierarchical *1–2–N* refinement: (1) node-wise feature selection refines the information available at each node; (2) sample-wise structure learning refines which node pairs constitute positives; and (N) a degree-aware focal loss reweights the one-to-many relationships between each anchor node and its neighborhood. Although feature selection is implemented as a preprocessing step, all three components are explicitly designed around the same neighborhood-based contrastive setting.

To this end, we propose Information-Refinement Graph Contrastive Learning (**IRGCL**), a *single-view framework* that integrates these three complementary modules into one cohesive pipeline: (1) *Feature selection* via low-rank approximation with Laplacian regularization, filtering out noisy attributes while preserving structure-consistent signals; (2) *Structure learning* via high-confidence clustering, reinforcing reliable intra-cluster edges while pruning spurious ones; (3) *Degree-aware focal contrast*, built on a Jensen–Shannon divergence (JSD) objective, dynamically balancing learning across low- and high-degree nodes. Together, these yield cleaner features, refined neighborhoods, and balanced contrastive signals for single-view GCL.

We evaluate IRGCL on transductive and inductive node classification, node clustering, and link prediction benchmarks. It consistently outperforms strong baselines, achieving noticeable improvements in challenging inductive settings such as PPI, and reduces degree bias in degree-stratified evaluations. The results demonstrate that jointly refining features, structure, and contrastive weighting enables effective single-view learning without fussy augmentations, while maintaining strong generalization across diverse graph tasks.

**Our contributions are threefold:**

- **Single-view information-refinement framework.** We introduce **IRGCL**, a principled single-view graph contrastive framework that jointly addresses three key bottlenecks for neighborhood-based contrast—feature noise, structural unreliability, and degree imbalance—through a unified 1–2–N refinement of nodes, positive pairs, and node–neighborhood interactions.
- **Theoretically grounded structure refinement.** We design a confidence-guided clustering and edge-editing rule with provable guarantees: the edits monotonically decrease Dirichlet energy and improve global homophily / conductance, providing a rigorous foundation for refining neighborhoods used as positives in the contrastive loss.
- **Comprehensive empirical validation.** Through extensive experiments on transductive and inductive node classification, node clustering, and link prediction, we show that **IRGCL** achieves state-of-the-art performance and substantially reduces degree bias. Ablation studies confirm that feature selection, structure learning, and degree-aware focal JSD act as complementary components, each critical to the observed gains.

## 2 RELATED WORKS

**Feature Selection.** Classical feature selection methods are categorized as *filter*, *wrapper*, and *embedded*. Filter methods (e.g., Relief (Kira & Rendell, 1992), Laplacian Score (He et al., 2005)) evaluate features via intrinsic statistics such as variance or smoothness, offering efficiency but ignoring model interactions. Wrapper methods (e.g., RFE (Guyon et al., 2002)) assess subsets by model performance, achieving accuracy at high computational cost. Embedded methods (e.g., Lasso, XG-Boost (Neyra et al., 2024)) incorporate selection during training via sparsity or importance scores. However, most traditional approaches fail to take into account topology when selecting features for graphs. Recent advances address this by combining low-rank approximation with structure-aware regularization Wang & Wang (2017). Our work builds on this idea by adapting low-rank approximation with Laplacian regularization specifically for graph contrastive learning settings, filtering noisy or redundant features while preserving topology-consistent signals.

**Graph Structure Learning.** Graph Structure Learning (GSL) refines graphs to better match features and tasks (Zhu et al., 2021a). Approaches include: (i) *metric-based*, constructing edges via similarity (e.g., AGCN (Ying et al., 2018), IDGL (Chen et al., 2020)); (ii) *neural*, predicting edge weights with deep models (e.g., PTDNet (Luo et al., 2021)); and (iii) *direct optimization*, learning adjacency with sparsity or smoothness constraints (e.g., BGCN (Zhang et al., 2019b)). Metric-based methods are efficient and interpretable but brittle under noise. We instead apply a *confidence filter* that refines topology by adding reliable intra-cluster edges while pruning spurious ones. As a result, safeguards on minimum degree and edit budgets prevent over-pruning, yielding graphs with improved homophily and reliability.

**Graph Contrastive Learning.** Graph contrastive learning (GCL) can be grouped into: (i) *augmentation-based* methods (e.g., DGI (Veličković et al., 2019), GraphCL (You et al., 2020), GRACE (Zhu et al., 2020b)); (ii) *non-augmentation* methods using latent perturbations or multiple encoders (e.g., SUGRL (Mo et al., 2022), SimGCL (Yu et al., 2022)); (iii) *input–latent* contrast (e.g., GMI (Peng et al., 2020b)); and (iv) *single-view* methods, which construct positives directly from neighborhoods (e.g., SIGNA (Sun et al., 2025)). Single-view approaches are efficient but still suffer from neighborhood bias and degree imbalance, which favors high-degree nodes and weakens low-degree ones (Hu et al., 2025). Our work advances this line by introducing a neighborhood- and degree-aware single-view contrastive strategy with degree-adaptive reweighting under a JSD objective, yielding balanced representations across sparse and dense regions.

## 3 METHODOLOGY

**Notations.** Let $G = (V, \mathcal{E})$ denote an undirected graph with node set $V = \{v_1, v_2, \ldots, v_N\}$ and edge set $\mathcal{E} \subseteq V \times V$. The node feature matrix is $X = [x_1, x_2, \ldots, x_N]^\top \in \mathbb{R}^{N \times d}$, where $x_i \in \mathbb{R}^d$ is the feature vector of node $v_i$. The adjacency matrix is $A \in \{0, 1\}^{N \times N}$ with $a_{ij} = 1$ if $(v_i, v_j) \in \mathcal{E}$ and 0 otherwise. We denote the learned node embeddings by $Z = [z_1, z_2, \ldots, z_N]^\top \in \mathbb{R}^{N \times d'}$. For each node $u$, its one-hop neighbors are $\mathcal{N}(u) = \{v \mid (u, v) \in \mathcal{E}\}$, its degree is $d_u = |\mathcal{N}(u)|$, and $D(V) = \{d_u\}_{u \in V}$ denotes the degree distribution with mean $\overline{D}(V)$ and standard deviation $\text{std}(D(V))$. Unless stated otherwise, the graph is unweighted ($w_{ij} = 1$ for $(i, j) \in \mathcal{E}$).

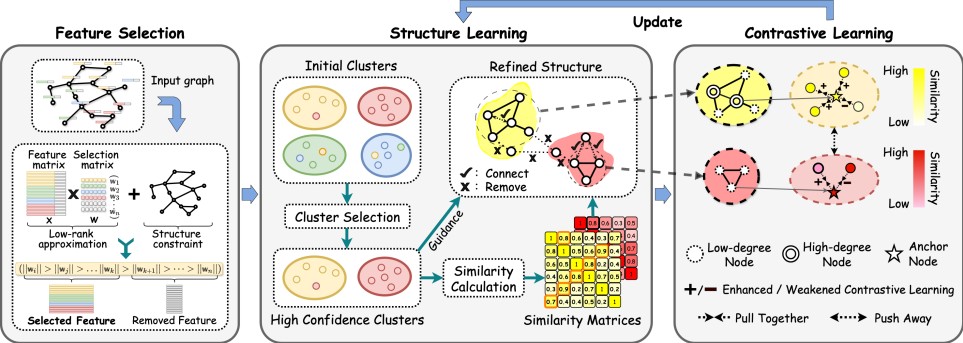

Figure 2: The framework of our proposed IRGCL.The method integrates three modules: (1) Feature Selection via low-rank approximation with structural regularization to retain informative, graph-consistent features; (2) Structure Learning via silhouette-guided clustering to reinforce reliable intra-cluster edges and prune noisy links; and (3) Contrastive Learning with degree-aware focal JSD to balance low- and high-degree nodes. Together, these yield cleaner features, refined topology, and balanced contrastive signals.

## 3.1 FEATURE SELECTION VIA LOW-RANK APPROXIMATION

A central obstacle in graph contrastive learning is that noisy or redundant attributes can distort similarity estimation and degrade representations. While classical low-rank models, such as non-negative matrix factorization (NMF) and singular value decomposition (SVD), are effective for de-noising and feature selection on i.i.d. data (Wang & Wang, 2017; Wang et al., 2015; Lai et al., 2023), they fail to exploit graph topology and thus overlook structural consistency. To address this, we adopt a *structure-aware low-rank feature selection* strategy that augments classical low-rank approximation with graph Laplacian regularization. This design highlights attributes aligned with local connectivity while suppressing noisy or topology-inconsistent signals.

Formally, the unsupervised low-rank approximation problems is:

$$\min_{W} \|X - XW\|_F^2 + \gamma \|W\|_{2,1}, \tag{1}$$

where $X \in \mathbb{R}^{n \times d}$ is the feature matrix and $W \in \mathbb{R}^{d \times d}$ the coefficient matrix. The mixed norm $\|W\|_{2,1} = \sum_{i=1}^{d} \|w_i\|_2$ (rows $w_i^\top$) promotes row sparsity and robustness; $\gamma > 0$ balances sparsity and reconstruction. Each $\|w_i\|_2$ serves as an importance score for feature $i$.

To integrate graph structure, we extend this formulation with a Laplacian regularizer and an explicit low-rank constraint:

$$\min_{W} \|X - XW\|_F^2 + \alpha \operatorname{Tr}(W^\top X^\top L X W) + \gamma \|W\|_{2,1} \quad \text{s.t. } \operatorname{rank}(W) = k, \tag{2}$$

where $L = D - A$ is the combinatorial Laplacian, $\alpha > 0$ controls topology smoothness, and $k$ is the desired number of retained features. Here $W$ acts as a projection matrix; enforcing $\operatorname{rank}(W) = k$ forces only $k$ dominant features to remain.

Following (Wang & Wang, 2017), the rank constraint is relaxed via Fy Fan's (Fan, 1949) theorem. Since $\operatorname{rank}(W) = \operatorname{rank}(WW^\top)$, we penalize the $d-k$ smallest eigenvalues:

$$\sum_{i=1}^{d-k} \sigma_i(WW^\top) = \min_{F \in \mathbb{R}^{d \times (d-k)}, \, F^\top F = I} \operatorname{Tr}(F^\top WW^\top F).$$

Substituting the above relaxations gives the final feature-selection objective

$$\min_{W,F} \mathcal{O}(W,F) = \frac{1}{2}\|X - XW\|_F^2 + \frac{\alpha}{2} \operatorname{Tr}(W^\top X^\top L X W) + \frac{\beta}{2} \operatorname{Tr}(W^\top F F^\top W) + \gamma \|W\|_{2,1}, \tag{3}$$

subject to $F^\top F = I$, where $X \in \mathbb{R}^{n \times d}$, $L = D - A$ is the graph Laplacian, and $\|W\|_{2,1} = \sum_{i=1}^{d} \|w_i\|_2$ promotes row sparsity so that only $k$ feature rows have large norms.

We optimize equation 3 by alternating updates in $F$ and $W$:

- **Update $F$.** For fixed $W$, $\text{Tr}(W^\top F F^\top W)$ is minimized when $F$ contains the eigenvectors of $WW^\top$ associated with its smallest eigenvalues; we take the $d-k$ smallest, concentrating the energy of $WW^\top$ in a rank-$k$ subspace.

- **Update $W$.** For fixed $F$, we majorize the non-smooth $\ell_{2,1}$ term by a quadratic surrogate using a diagonal reweighting matrix $\Gamma^{(t)} = \text{diag}(g_1^{(t)}, \ldots, g_d^{(t)})$ with $g_i^{(t)} = 1/(\|w_i^{(t)}\|_2 + \varepsilon)$. This yields a convex quadratic surrogate $\mathcal{Q}(W \mid W^{(t)}, F^{(t)})$ whose unique minimizer is

$$W^{(t+1)} = \left( X^\top X + \alpha X^\top L X + \beta F^{(t)}(F^{(t)})^\top + \gamma \Gamma^{(t)} \right)^{-1} X^\top X.$$

By construction, $\mathcal{Q}$ upper-bounds $\mathcal{O}(\cdot, F^{(t)})$ and is tight at $W^{(t)}$, so $\mathcal{O}(W^{(t+1)}, F^{(t)}) \leq \mathcal{O}(W^{(t)}, F^{(t)})$ and the objective decreases monotonically. Since $\mathcal{O}$ is bounded below, the sequence converges, and standard majorization–minimization theory implies that every limit point of $\{W^{(t)}\}$ is a stationary point of $\mathcal{O}(W, F)$. The detailed proof is provided in the appendix. Finally, we select the $k$ features corresponding to the rows of $W$ with the largest $\|w_i\|_2$. In the unified 1–2–N design of IRGCL, this module plays the *node-wise (level-1)* role: it refines each node's attribute vector so that the encoder and similarity function $D_\phi(u, v)$ used in the contrastive loss operate on denoised, structure-aligned features. This prepares a cleaner similarity landscape for the subsequent pair-wise structure refinement and degree-aware contrastive reweighting.

## 3.2 STRUCTURE LEARNING VIA HIGH-CONFIDENCE CLUSTERING

Directly sampling contrastive pairs from the original structure can be risky, which motivates us to refine the graph topology. To this end, we iteratively refine the topology using reliable clusters. Specifically, every $T$ epochs, we run $K$-means on embeddings $\mathbf{H}$ to obtain clusters $\{c_k\}_{k=1}^C$ and then compute node silhouettes. A cluster $c_k$ is deemed *high-confidence* if its local average silhouette score exceeds the global average, i.e., $\bar{S}_k = |c_k|^{-1} \sum_{i \in c_k} s_i > \bar{S} = N^{-1} \sum_{i=1}^N s_i$, where $s_i = \frac{b_i - a_i}{\max\{a_i, b_i\}}$ is the silhouette score of node $i$, $a_i$ is its average intra-cluster distance, $b_i$ is its smallest average distance to another cluster, $\bar{S}_k$ is the average silhouette within cluster $c_k$, and $\bar{S}$ is the graph-level average silhouette.

Under this edit policy, we call an edge *spurious* if (i) it connects nodes assigned to different high-confidence clusters (cross-cluster edge), or (ii) it is an intra-cluster edge within a high-confidence cluster whose silhouette-scaled similarity (defined below) is below the threshold $\tau$.

**Silhouette-scaled similarity.** Given a high-confidence cluster, the similarity between any two nodes $i$ and $j$ in the cluster is computed from their representations $\mathbf{Z}$ as

$$\text{Sim}(i, j) = \max\left\{0, \, 1 - \frac{\|\mathbf{Z}_i - \mathbf{Z}_j\|}{2(s_i + 1)(s_j + 1)}\right\}. \tag{4}$$

In this way, pairs with high silhouette scores are prioritized and receive higher similarity.

**Edit policy (node-local).** For each node $i$ in any high-confidence cluster: *(a)* add an edge between node $i$ and its most similar non-connected intra-cluster node

$$j^\star = \arg\max_{j \in c(i), \, (i,j) \notin \mathcal{E}} \text{Sim}(i, j);$$

*(b)* remove the weakest intra-cluster edge if any $\text{Sim}(i, j) < \tau$ (default $\tau = 0.9$):

$$j^\flat = \arg\min_{j \in \mathcal{N}(i) \cap c(i)} \text{Sim}(i, j);$$

*(c)* remove inter-cluster edges $(i, j)$ if nodes $i$ and $j$ are not in the same cluster. After edits, the graph is symmetrized, and a minimum-degree safeguard is enforced (a concise description of this safeguard is provided in the appendix). From the contrastive-learning viewpoint, this structure-learning module performs *sample-wise* (pair-level) refinement of positives and negatives: by adding high-confidence intra-cluster edges and pruning spurious ones, it reshapes the one-hop neighborhoods $P_u = \mathcal{N}(u)$ on which our single-view contrastive loss is defined. In the unified $1-2-N$ hierarchy, it plays the level-2 "pair" role between node-wise feature refinement and node–neighborhood reweighting.

**Mechanism and guarantees.** Let $L$ be the base Laplacian and $L'$ be the edited one. For any $f \in \mathbb{R}^N$, the Dirichlet energy is $\Phi_L(f) = f^\top L f = \sum_{(i,j)\in\mathcal{E}} w_{ij}(f_i - f_j)^2$. Then, we have:

**Theorem 3.1** (Monotone Dirichlet energy). *Fix a base graph $G$ and one round of confidence-guided edits to obtain $G'$. If $f$ is (approximately) clusterwise-constant on the high-confidence clusters, then $\Phi_{L'}(f) \leq \Phi_L(f)$, with strict inequality if any inter-cluster edge is removed.*

*Interpretation.* Intra-cluster additions keep $\Phi$ unchanged when $f_i = f_j$; removing the weakest intra-cluster edge also preserves $\Phi$; deleting inter-cluster edges reduces $\sum w_{ij}(f_i - f_j)^2$ for $f_i \neq f_j$. Thus cluster-consistent signals are pushed to lower graph frequencies.

**Corollary 3.2** (Cut/conductance and homophily). *For any high-confidence cluster $c$, $\mathrm{Ncut}_G(c)$ does not increase (conductance improves), where $\mathrm{Ncut}_G(c)$ measures the fraction of edges leaving the cluster. Smaller $\mathrm{Ncut}_G(c)$ means better within-cluster cohesion (high conductance). Consequently, the average intra-cluster affinity (global homophily) increases monotonically.*

**Spectral view.** For a (soft) indicator $h_C$, the Rayleigh quotient $\mathcal{R}_L(h_C) = \frac{h_C^\top L h_C}{\|h_C\|^2}$ decreases (Theorem 3.1), pushing cluster directions toward low eigenvalues that GNNs/filters amplify; Coupled with Cheeger-type Chung (1997); von Luxburg (2007); Shi & Malik (2000) relations that link cuts to spectrum, this explains the empirical rise of homophily and stronger linear probes with fewer layers.

**Deployment modes.** *Rebase-to-$G_0$:* edit the fixed $G_0$ each round to get $G'_t$ (used for sampling/-contrast); then $\Phi_{L'_t}(f) \leq \Phi_{L_0}(f)$ for every $t$, while the message passing (MP) graph stays fixed. *Carry-forward:* edit $G_{t-1} \to G_t$; then $\Phi_{L_t}(f) \leq \Phi_{L_{t-1}}(f)$. This maximizes smoothing but may over-prune heterophilous edges. We default to *rebase-to-$G_0$ for edits, fixed $G_0$ for MP*.

**Why not random masking (vs. SIGNA).** Random neighbor masking can drop informative intra-cluster edges, gives no monotone energy guarantee, and increases variance; our confidence-guided edits provably reduce energy and cross-cluster cuts, stabilizing alignment while preserving intra-cluster connectivity.

**Practical notes.** Edits occur every $T$ epochs with degree safeguards. We track (i) global homophily, (ii) per-cluster Rayleigh quotients of $h_C$, and (iii) $\mathrm{Ncut}(C)$, which empirically confirm *monotonic trends*. Full proofs appear in Appendix.

### 3.3 CONTRASTIVE LEARNING WITH NEIGHBORHOOD & DEGREE AWARENESS

Contrastive learning on graphs often assumes abundant negatives and balanced neighborhoods, but real graphs rarely satisfy these conditions: one-hop neighborhoods can become noisy or imbalanced after pruning, while low-degree nodes lack sufficient positives and high-degree nodes contain overly mixed ones. For node $u$, positives $P_u = \mathcal{N}(u)$, where $\mathcal{N}(u)$ is the one-hop neighbors of node $u$, negatives $N_u = V \setminus (P_u \cup \{u\})$. One-hop neighbors are restricted to positives, which shrinks the negative pool and makes InfoNCE and Donsker–Varadhan (DV) (v. d. Oord et al., 2018; Donsker & Varadhan, 1983; Hjelm et al., 2018) unstable. Motivated by the robustness of Jensen–Shannon (JSD)-based objectives under limited or biased negatives (Nowozin et al., 2016) and prior work on debiased / hardness-aware contrastive learning (Chuang et al., 2020; Robinson et al., 2021), we adopt the JSD estimator:

$$\mathbb{JSD}(u) = \mathbb{E}_{v^+ \sim P_u}\big[\log D_\phi(u, v^+)\big] + \mathbb{E}_{v^- \sim N_u}\big[\log(1 - D_\phi(u, v^-))\big], \tag{5}$$

with similarity discriminator $D_\phi : \mathbb{R}^d \times \mathbb{R}^d \to [0, 1]$.

We use normalized cosine similarity calculated on top of embeddings $z_u \in \mathbb{R}^d$:

$$D_\phi^{\mathrm{norm}}(u, v) = \frac{\cos(z_u, z_v) + 1}{2}, \quad \cos(z_u, z_v) = \frac{z_u^\top z_v}{\|z_u\|_2 \|z_v\|_2}. \tag{6}$$

Inspired by focal loss for hard-example mining in detection (Lin et al., 2017) and hardness-aware contrastive objectives (Chuang et al., 2020; Robinson et al., 2021), we introduce a degree-adaptive focal weighting on positive terms. Let $s_v = D_\phi^{\mathrm{norm}}(u, v)$. We define a numerically stable softmax:

$$\mathrm{Softmax}_{v \in P_u}(s_v) = \frac{\exp\big(s_v - \max_{v'} s_{v'}\big)}{\sum_{v'} \exp\big(s_{v'} - \max_{v''} s_{v''}\big)}, \quad w^{\mathrm{focal}}(u, v^+) = \big(1 - \mathrm{Softmax}_{v \in P_u}(s_v)\big)^{\gamma^u}, \tag{7}$$

where $\gamma^u = (\overline{D}(V) - d_u)/(\text{std}(D(V)))$ is the degree-adaptive exponent that is clipped to $[-1, 1]$ to avoid instability and $\gamma^u$ is computed deterministically from degrees (no extra hyperparameter), $d_u = |\mathcal{N}(u)|$ is the degree of node $u$, and $\overline{D}(V)$ and $\text{std}(D(V))$ are the mean and standard deviation of the degree distribution. This design is also informed by recent analyses of degree bias in graph contrastive learning (Hu et al., 2025), which show that low- and high-degree nodes behave very differently under standard objectives.

For low-degree nodes ($\gamma^u > 0$), harder positives (low similarity) receive larger weights; for high-degree nodes ($\gamma^u < 0$), low-similarity edges are downweighted to suppress noisy positives. From the unified 1–2–N view, this degree-aware focal JSD plays the level-$N$ role: for each anchor $u$, it reweights the one-to-many interactions between $u$ and all neighbors in $P_u$, amplifying informative positives for sparse nodes while tempering noisy positives for hubs and thereby reshaping the node–neighborhood contrastive loss.

So the final objective loss is formulated as:

$$\ell(u) = -\frac{1}{|P_u|} \sum_{v^+ \in P_u} w^{\text{focal}}(u, v^+) \log D_\phi^{\text{norm}}(u, v^+) - \frac{1}{|N_u|} \sum_{v^- \in N_u} \log\left(1 - D_\phi^{\text{norm}}(u, v^-)\right), \quad (8)$$

$$\mathcal{L} = \frac{1}{|V|} \sum_{u \in V} \ell(u). \quad (9)$$

## 4 EXPERIMENT

### 4.1 EXPERIMENT SETUP

**Datasets.** We comprehensively evaluate **IRGCL** on a diverse collection of benchmark graphs, covering transductive, inductive, and multi-graph learning settings. These include citation networks Cora, Citeseer, and Pubmed (Yang et al., 2016), the WikiCS (Mernyei & Cangea, 2020), product co-purchase graphs Amazon Photo and Amazon Computers (Shchur et al., 2018), and the co-authorship network **Coauthor CS** (Shchur et al., 2018). To test inductive generalization, we use the **PPI** dataset (Zitnik & Leskovec, 2017), a collection of protein–protein interaction networks where each graph corresponds to a distinct human tissue. These datasets vary widely in size, feature dimension, homophily, and degree distribution, providing a comprehensive basis for evaluating transductive and inductive performance.

**Baselines.** We compare **IRGCL** with representative methods across five categories: supervised GNNs (GCN (Kipf & Welling, 2017), GAT (Veličković et al., 2018)) as fully supervised upper bounds; augmentation-based contrastive methods (DGI (Veličković et al., 2019), MVGRL (Hassani & Khasahmadi, 2020), GRACE (Zhu et al., 2020b), GCA (Zhu et al., 2021b), BGRL (Thakoor et al., 2021)); input–latent contrast (GMI (Peng et al., 2020b)); dual-encoder contrast (SUGRL (Mo et al., 2022), AFGRL (Lee et al., 2021), PolyGCL (Chen et al., 2024)); and single-view contrast (GIC (Mavromatis & Karypis, 2020), SIGNA (Sun et al., 2025), and our **IRGCL**). For link prediction, we additionally consider structure-focused or heterophily-aware models such as $H^2$GCN (Zhu et al., 2020a), GPR-GNN (Chien et al., 2021), SLIMG (Yoo et al., 2023), and NETINFOF (Lee et al., 2024), alongside standard MPNNs (GraphSAGE, GAT).

| Method | Train Data | F1-score |
|---|---|---|
| GaAN-mean | $\mathbf{X}, \mathbf{A}, \mathbf{Y}$ | 96.90±0.20 |
| GAT | $\mathbf{X}, \mathbf{A}, \mathbf{Y}$ | 97.30±0.20 |
| FastGCN | $\mathbf{X}, \mathbf{A}, \mathbf{Y}$ | 63.70±0.60 |
| Unsup-GraphSAGE | $\mathbf{X}, \mathbf{A}$ | 46.50 |
| Random-Init | $\mathbf{X}, \mathbf{A}$ | 62.60±0.20 |
| DGI | $\mathbf{X}, \mathbf{A}$ | 63.80±0.20 |
| GMI | $\mathbf{X}, \mathbf{A}$ | 64.60±0.00 |
| S$^2$GRL | $\mathbf{X}, \mathbf{A}$ | 66.00±0.00 |
| GRACE | $\mathbf{X}, \mathbf{A}$ | 66.20±0.10 |
| Subg-Con | $\mathbf{X}, \mathbf{A}$ | 66.90±0.20 |
| GraphCL-NS | $\mathbf{X}, \mathbf{A}$ | 65.90±0.00 |
| BGRL-GAT-Enc. | $\mathbf{X}, \mathbf{A}$ | 70.49±0.05 |
| SIGNA | $\mathbf{X}, \mathbf{A}$ | 92.25±0.03 |
| **IRGCL** | $\mathbf{X}, \mathbf{A}$ | **93.05±0.05** |

Table 1: PPI inductive node classification (micro-F1).

**Evaluation Protocols.** We train all models in a fully unsupervised manner and then freeze the encoder for downstream evaluation. More details can be found in appendix.

| | Method | Cora | Citeseer | Pubmed | Amazon Photo | Amazon Computers |
|---|---|---|---|---|---|---|
| Supervised methods | GCN | 81.50±0.20 | 70.30±0.40 | 79.00±0.50 | 92.42±0.22 | 86.51±0.54 |
| | GAT | 83.00±0.20 | 72.50±0.30 | 79.00±0.50 | 92.56±0.35 | 86.93±0.29 |
| Aug-based cross-view | DGI | 82.30±0.50 | 71.50±0.40 | 76.80±0.60 | 91.61±0.22 | 83.95±0.47 |
| | MVGRL | 82.90±0.30 | 72.60±0.40 | 80.10±0.70 | 92.08±0.01 | 87.52±0.21 |
| | GRACE | 83.10±0.20 | 72.10±0.10 | 80.60±0.40 | 92.24±0.45 | 86.35±0.44 |
| | BGRL | 82.70±0.60 | 71.10±0.80 | 79.60±0.50 | 92.87±0.27 | 89.68±0.31 |
| Input-latent cross-view | GMI | 82.40±0.60 | 72.40±0.20 | 79.90±0.40 | 90.68±0.17 | 82.21±0.31 |
| Dual-encoder cross-view | SUGRL | 83.40±0.50 | 73.00±0.40 | 81.90±0.30 | 93.07±0.15 | 88.93±0.21 |
| | AFGRL | 81.30±0.20 | 68.70±0.30 | 80.60±0.40 | 93.22±0.28 | 89.88±0.33 |
| | PolyGCL | 83.34±0.50 | **74.50±0.30** | 82.40±0.30 | 93.50±0.50 | 89.70±0.70 |
| Single-view | GIC | 81.70±0.80 | 71.90±0.90 | 77.40±0.50 | 91.60±0.10 | 84.90±0.20 |
| | SIGNA | 81.90±0.40 | 71.00±0.10 | 81.30±0.70 | 95.32±0.19 | 90.46±0.25 |
| | **IRGCL** | **83.70±0.40** | 73.20±0.20 | **82.50±0.40** | **95.40±0.18** | **90.50±0.22** |

Table 2: Performance on transductive node classification tasks. The best and second-best performances among unsupervised methods are highlighted in bold and underlined, respectively.

## 4.2 EXPERIMENTAL RESULTS

**Inductive node classification.** Table 1 reports results on the multi-graph PPI benchmark in terms of micro-averaged F1. As expected, supervised GNNs such as GAT and GaAN achieve the strongest overall scores (97.3% and 96.9%), serving as an upper bound.

In contrast, most unsupervised baselines perform poorly, with F1 ranging between 46–70% (e.g., Unsup-GraphSAGE at 46.5%, GRACE at 66.2%). Notably, BGRL with a GAT encoder reaches 70.5%, but still falls far behind supervised counterparts. Recent single-view methods close this gap substantially: SIGNA achieves 92.25%, a remarkable improvement over prior unsupervised models. Our method further advances the state of the art, reaching 93.05% F1, which is both higher than SIGNA and dramatically beyond all other unsupervised baselines. This performance underscores the strength of our unified framework in inductive scenarios, where generalization across disjoint graphs is particularly challenging.

**Transductive node classification.** The empirical results across five benchmark datasets are summarized in Table 2. Our method attains the best or second-best performance among all unsupervised competitors on every dataset, achieving new state-of-the-art single-view results on four out of five benchmarks (Cora, PubMed, Amazon Photo, and Amazon Computers), and ranking second only to the dual-encoder PolyGCL on Citeseer. Compared with the previous single-view state-of-the-art SIGNA, IRGCL improves accuracy by +1.8, +2.2, +1.2, +0.08, and +0.04 percentage points on Cora, Citeseer, PubMed, Amazon Photo, and Amazon Computers, respectively. It also consistently surpasses the strong dual-encoder SUGRL across all datasets (e.g., by +0.3, +0.2, and +0.6 percentage points on Cora, Citeseer, and PubMed). These gains, though modest on some datasets and larger on others, are consistently observed across diverse graph types, showing that integrating feature selection, structure refinement, and degree-aware contrast yields robust benefits beyond SIGNA's random neighbor masking. Moreover, while several dual-encoder methods (e.g., AFGRL and SUGRL) exhibit more pronounced fluctuations across datasets, our single-view approach delivers stable leading performance on all five transductive benchmarks, underscoring its practical reliability.

**Node clustering.** We assess representation quality via unsupervised K-means clustering, reporting Normalized Mutual Information (NMI) and Homogeneity (Table 3). IRGCL achieves the highest scores across all datasets; for example, on WikiCS it improves NMI to 0.5161 (+12% over SIGNA's 0.4593), with similar gains on Amazon Photo (0.7799

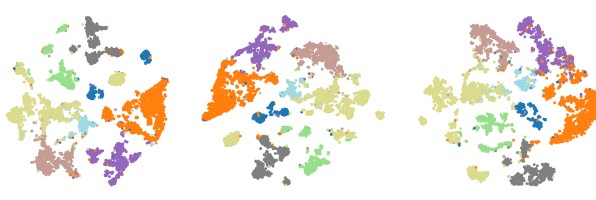

Figure 3: t-SNE visualizations of node embeddings: **IRGCL** (left), SIGNA (middle), SUGRL (right).

vs. 0.7635) and Amazon Computers (0.5832 vs. 0.5608). Even on the strong Coauthor CS baseline, it slightly improves performance (0.8084 vs. 0.8047). These results show that IRGCL produces

| Method | Wiki CS | | Am. Photo | | Am. Computers | | Co. CS | |
|---|---|---|---|---|---|---|---|---|
| | NMI | Homo. | NMI | Homo. | NMI | Homo. | NMI | Homo. |
| Raw features | 0.2633 | 0.2738 | 0.3273 | 0.3376 | 0.2389 | 0.2617 | 0.7103 | 0.7446 |
| GRACE | 0.4282 | 0.4423 | 0.6513 | 0.6657 | 0.4793 | 0.5222 | 0.7562 | 0.7909 |
| GCA | 0.3373 | 0.3525 | 0.6443 | 0.6575 | 0.5278 | 0.5816 | 0.7620 | 0.7965 |
| BGRL | 0.3969 | 0.4156 | 0.6841 | 0.7004 | 0.5364 | 0.5869 | 0.7732 | 0.8041 |
| AFGRL | 0.4132 | 0.4307 | 0.6563 | 0.6743 | 0.5520 | 0.6040 | 0.7859 | 0.8161 |
| SIGNA | 0.4593 | 0.4763 | 0.7635 | 0.7823 | 0.5608 | 0.6057 | 0.8047 | 0.8408 |
| **IRGCL** | **0.5161** | **0.5239** | **0.7799** | **0.8023** | **0.5832** | **0.6413** | **0.8084** | **0.8442** |

Table 3: Performance on node clustering tasks in terms of NMI and Homogeneity.

| Method | Cora | Citeseer | PubMed | Am.Photo | Am.Computers |
|---|---|---|---|---|---|
| GCN | 67.10±1.80 | 60.40±10.00 | 47.60±13.00 | 39.10±1.60 | 22.50±3.10 |
| SAGE | 68.40±2.80 | 55.90±2.50 | 57.60±1.10 | 40.00±1.90 | 27.50±2.10 |
| GAT | 66.70±3.60 | 65.20±2.60 | 55.10±2.40 | 44.20±3.50 | 28.30±1.60 |
| $H^2$GCN | 64.40±3.40 | 35.70±5.40 | 50.50±0.90 | 29.50±2.40 | 17.90±0.70 |
| GPR-GNN | 69.80±1.90 | 53.50±8.10 | 66.30±3.30 | 34.10±1.10 | 20.70±1.80 |
| SLIMG | 77.90±1.30 | 86.80±1.00 | 55.90±2.80 | 40.20±2.50 | 25.30±0.90 |
| NETINFOF | 81.30±0.60 | 87.30±1.30 | 59.70±1.10 | 46.80±2.20 | 31.10±1.90 |
| SIGNA | 94.33±0.67 | 94.19±0.20 | 77.28±2.10 | 45.21±1.57 | 27.87±0.80 |
| **IRGCL** | **96.55±0.96** | **98.81±0.29** | **78.79±2.52** | **51.21±0.81** | **31.54±2.82** |

Table 4: Homogeneous graph link prediction evaluated with Hits@100 (Lee et al., 2024).

embeddings that are not only discriminative but also inherently more clusterable. Figure 3 further illustrates this: compared with SIGNA and SUGRL, `IRGCL` yields more compact, well-separated t-SNE clusters, highlighting its ability to capture semantically coherent structures.

**Link prediction.** Table 4 reports Hits@100 results on homogeneous link prediction following the NetInfoF protocol (Lee et al., 2024). Traditional GNNs and heterophily-aware models perform poorly, while contrastive approaches provide large gains. SIGNA achieves strong results on citation datasets, but our method consistently surpasses it, reaching 96.6 on Cora, 98.8 on Citeseer, and 78.8 on PubMed. On product graphs, we also outperform NETINFOF, achieving 51.2 on Amazon Photo and 31.5 on Amazon Computers. These results confirm that our unified framework yields embeddings that generalize well for link prediction, even against specialized baselines.

## 4.3 ABLATION STUDY

IRGCL consists of three core components—feature denoising, structure refinement, and degree-aware contrastive learning. To verify their contributions, we evaluate: (1) the effect of structure-aware low-rank feature selection, (2) the benefits of high-confidence clustering–based structure learning, and (3) the effectiveness of the norm-JSD loss combined with degree-aware focal reweighting.

**Effectiveness of the feature selection.** We evaluate our low-rank, Laplacian-regularized feature selector by varying the retained feature ratio from 0.05 to 1.0 across five datasets (Figure. 4). All datasets benefit from removing redundant features. Amazon Photo shows a steep rise from about 60% to over 90% accuracy as the ratio increases, peaking near 0.9. Amazon Computers exhibits the largest relative improvement, climbing from around 50% to around

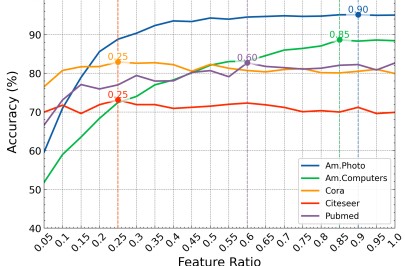

Figure 4: Comparison of feature ratio effects on accuracy across datasets.

85%, and remaining stable thereafter. Cora improves sharply and stabilizes after only 25% of features are kept, indicating that a small, clean subset suffices. Citeseer peaks early around 25% and then remains flat, suggesting heavy redundancy in its original features Pubmed reaches its highest

| Variant | Am. Photo | Am. Computers | Cora | Citeseer | Pubmed |
|---|---|---|---|---|---|
| **IRGCL** | 95.40±0.18 | 90.50±0.22 | 83.70±0.40 | 73.20±0.20 | 82.50±0.40 |
| Include All Clusters | 94.04±0.26 | **OOM** | 81.20±0.14 | 70.50±0.07 | 80.40±0.10 |
| Without Structural Learning | 95.04±0.29 | 90.12±0.23 | 81.90±0.28 | 70.60±0.20 | 81.20±0.07 |

Table 5: Ablation study of high-confidence clustering structure learning

| Variant | Am. Photo | Am. Computers | Cora | Citeseer | Pubmed |
|---|---|---|---|---|---|
| Norm-JSD with DAF | **95.40 ± 0.18** | **90.50 ± 0.22** | **83.70 ± 0.40** | **73.20 ± 0.20** | **82.50 ± 0.40** |
| Norm-JSD without DAF | 95.25 ± 0.23 | 90.14 ± 0.25 | 81.50 ± 0.50 | 71.75 ± 0.40 | 81.10 ± 0.10 |
| InfoMax with DAF | 95.09 ± 0.21 | 89.77 ± 0.42 | 81.34 ± 0.65 | 70.76 ± 0.05 | 80.98 ± 0.04 |
| InfoMax without DAF | 95.04 ± 0.16 | 89.34 ± 0.29 | 80.96 ± 0.43 | 70.18 ± 0.13 | 80.54 ± 0.25 |

Table 6: Ablation study of different objectives

performance near 60% and stays stable thereafter. Overall, these results confirm that our structure-aware feature selection effectively filters noise, preserves essential information, and improves down-stream classification across diverse graph domains.

**Effectiveness of the structure learning via high-confidence clustering.** We evaluate the impact of our silhouette-guided structural refinement by comparing three settings: the full `IRGCL` model, a variant that includes *all* clusters (ignoring the confidence filter), and a version with *no* structural learning. Table 5 shows that removing the confidence filter and using all clusters consistently harms accuracy. For example, performance drops from 95.40% to 94.04% on Amazon Photo and from 83.70% to 81.20% on Cora; the Amazon Computers variant fails with out-of-memory (OOM), showing the high computational cost of processing unreliable clusters. Completely disabling structural learning also reduces accuracy on every dataset (e.g., 95.40% → 95.04% on Photo, 83.70% → 81.90% on Cora, and 82.50% → 81.20% on Pubmed). These results confirm that our high-confidence filtering is both effective and efficient: it avoids the cost and instability of editing with noisy clusters while significantly improving the quality of the refined graph topology.

**Effectiveness of the degree-aware focal (DAF) norm-JSD.** Table 6 shows that norm-JSD with DAF achieves the best accuracy on all five datasets (e.g., 95.40% on Amazon Photo, 90.50% on Computers, 83.70% on Cora), outperforming the strongest InfoMax variant by about 1–3 percentage points, while plain norm-JSD already matches or exceeds InfoMax in most cases.

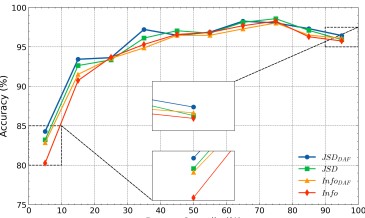

Figure 5 further stratifies performance by degree on Amazon Photo: although all objectives improve with degree, the DAF variant of norm-JSD yields the largest gains for both low- and high-degree nodes, indicating that degree-aware focal reweighting not only boosts overall accuracy but also mitigates degree-induced imbalance in representation quality.

Figure 5: Accuracy on Amazon Photo with four loss functions.

## 5 CONCLUSIONS

In this work, we presented **IRGCL**, a single-view graph contrastive learning framework that tackles three core challenges in unsupervised graph representation learning: noisy features, unreliable structures, and degree imbalance. IRGCL follows a hierarchical *1–2–N* refinement scheme: level-1 node-wise feature selection denoises attributes, level-2 sample-wise structure learning refines positive pairs via high-confidence edge edits, and level-$N$ degree-aware focal contrast reweights node–neighborhood losses. Together, these components yield robust and generalizable representations. Experiments across diverse benchmarks show consistent gains over strong baselines, and ablations confirm that each module contributes a distinct and complementary benefit.

ETHICS STATEMENT

Our work follows the ICLR Code of Ethics. We use only publicly available graph datasets (Cora, Citeseer, PubMed, Amazon Photo/Computers, etc.) with no personally identifiable information. No human subjects or sensitive personal data are involved, and our method poses no known safety, security, or environmental risks. We acknowledge that graph representation learning can potentially be misused for profiling or surveillance if applied to sensitive social graphs, and we encourage responsible use.

REPRODUCIBILITY STATEMENT.

We have made extensive efforts to ensure the reproducibility of our work.

**Datasets.** All datasets used in our experiments are publicly available and described in the experiments. 4.1.

**Encoder implementation.** On all tasks, our encoder consists of two encoding blocks (L = 2). For transductive node classification, node clustering and link prediction, we use the simple linear layer or graph convolutionial layer as the BaseEncoder, while for inductive node classification, the graph convolutionial layer is adopted for better inference effect.

**Code.** An anonymous link to our source code and configuration files is provided in the Abstract.

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

## A  THE USE OF LARGE LANGUAGE MODELS (LLMS)

Large Language Models (LLMs) were used to aid in grammar checking and typo elimination. In particular, we leveraged an LLM to refine language usage, enhance readability, and improve clarity throughout the paper. The model assisted with sentence rephrasing, grammatical correction, and improving the overall flow and coherence of the text.

It is important to note that the LLM was not involved in the ideation, research methodology, or experimental design. All research concepts, ideas, and analyses were developed and conducted by the authors. The contributions of the LLM were solely focused on improving the linguistic quality of the paper, with no involvement in the scientific content or data analysis.

The authors take full responsibility for the content of the manuscript, including any text generated or polished by the LLM. We have ensured that the LLM-generated text adheres to ethical guidelines and does not contribute to plagiarism or scientific misconduct.

## B  ALTERNATING UPDATES AND CONVERGENCE OF $W$.

The joint problem in $(W, F)$ is not globally convex, so alternating updates in $(W, F)$ need not reach the global optimum. This is standard for low-rank feature-selection formulations. Our aim is instead to show that, for fixed $F$, the subproblem in $W$ is convex and that the resulting update yields a stable procedure whose limit points are stationary for the feature-selection objective.

Up to constants independent of $W$ and $F$, the objective is

$$\mathcal{O}(W, F) = \frac{1}{2}\|X - XW\|_F^2 + \frac{\alpha}{2}\operatorname{Tr}\left(W^\top X^\top LXW\right) + \frac{\beta}{2}\operatorname{Tr}\left(W^\top FF^\top W\right) + \gamma\|W\|_{2,1}, \quad (10)$$

where $X \in \mathbb{R}^{n \times d}$, $L = D - A$ is the graph Laplacian, and $\|W\|_{2,1} = \sum_{i=1}^d \|w_i\|_2$. Since $L$ is symmetric positive semidefinite, for any fixed $F$ the matrix

$$M(F) := X^\top X + \alpha X^\top LX + \beta FF^\top \quad (11)$$

is symmetric positive semidefinite, and the smooth part of $\mathcal{O}(W, F)$ is a convex quadratic in $W$.

To handle the non-smooth $\ell_{2,1}$ term we use a standard majorization–minimization procedure based on iteratively reweighted least squares. At iteration $t$, define the diagonal matrix $\Gamma^{(t)} = \operatorname{diag}(g_1^{(t)}, \ldots, g_d^{(t)})$ with

$$g_i^{(t)} = \frac{1}{\|w_i^{(t)}\|_2} \quad \text{for all rows with } \|w_i^{(t)}\|_2 \neq 0, \quad (12)$$

and use the inequality $\|v\|_2 \le \frac{1}{2}\|v\|_2^2/\|v^{(t)}\|_2 + \frac{1}{2}\|v^{(t)}\|_2$ to majorize

$$\|W\|_{2,1} = \sum_{i=1}^d \|w_i\|_2 \le \frac{1}{2}\sum_{i=1}^d \frac{1}{\|w_i^{(t)}\|_2}\|w_i\|_2^2 + C^{(t)} = \frac{1}{2}\operatorname{Tr}\left(W^\top \Gamma^{(t)} W\right) + C^{(t)}, \quad (13)$$

for a constant $C^{(t)}$ independent of $W$.

Substituting this bound into equation 10 gives the quadratic surrogate

$$\mathcal{Q}(W \mid W^{(t)}, F^{(t)}) = \frac{1}{2}\|X - XW\|_F^2 + \frac{\alpha}{2}\operatorname{Tr}\left(W^\top X^\top LXW\right)$$
$$+ \frac{\beta}{2}\operatorname{Tr}\left(W^\top F^{(t)}(F^{(t)})^\top W\right) + \frac{\gamma}{2}\operatorname{Tr}\left(W^\top \Gamma^{(t)} W\right), \quad (14)$$

which satisfies

$$\mathcal{O}(W, F^{(t)}) \le \mathcal{Q}(W \mid W^{(t)}, F^{(t)}) + \operatorname{const}^{(t)},$$
$$\mathcal{O}(W^{(t)}, F^{(t)}) = \mathcal{Q}(W^{(t)} \mid W^{(t)}, F^{(t)}) + \operatorname{const}^{(t)}. \quad (15)$$

The gradient and Hessian of $\mathcal{Q}$ with respect to $W$ are

$$\frac{\partial \mathcal{Q}}{\partial W} = -X^\top X + \left(M(F^{(t)}) + \gamma\Gamma^{(t)}\right)W, \quad \frac{\partial^2 \mathcal{Q}}{\partial W^2} = M(F^{(t)}) + \gamma\Gamma^{(t)} \succeq 0, \quad (16)$$

so $\mathcal{Q}$ is a convex quadratic with unique minimizer

$$W^{(t+1)} = \left(M(F^{(t)}) + \gamma\Gamma^{(t)}\right)^{-1} X^\top X. \tag{17}$$

By construction, $\mathcal{Q}$ is a tight upper bound (majorizer) of $\mathcal{O}(\cdot, F^{(t)})$ at $W^{(t)}$, hence

$$\mathcal{O}(W^{(t+1)}, F^{(t)}) \leq \mathcal{Q}(W^{(t+1)} \mid W^{(t)}, F^{(t)}) \leq \mathcal{Q}(W^{(t)} \mid W^{(t)}, F^{(t)}) = \mathcal{O}(W^{(t)}, F^{(t)}), \tag{18}$$

so the sequence of objective values $\{\mathcal{O}(W^{(t)}, F^{(t)})\}_t$ is monotonically non-increasing. Since $\mathcal{O}$ is bounded below, these values converge, and standard majorization–minimization theory implies that every limit point of $\{W^{(t)}\}$ is a stationary point of $\mathcal{O}(W, F)$. In our experiments, different random initializations lead to very similar $W$ and downstream performance, suggesting that the procedure is stable in practice.

## C  HIGH-CONFIDENCE CLUSTERING FOR STRUCTURE LEARNING: MONOTONIC ENERGY AND HOMOPHILY IMPROVEMENT

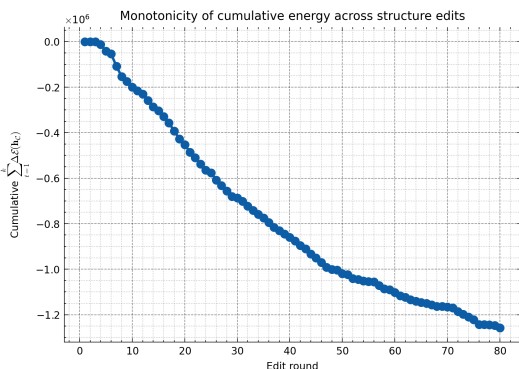

Figure 6: Cumulative energy decrease $\sum_C \Delta\Phi(h_C)$ across structure edit rounds. The monotonic decline confirms Theorem 3.1.

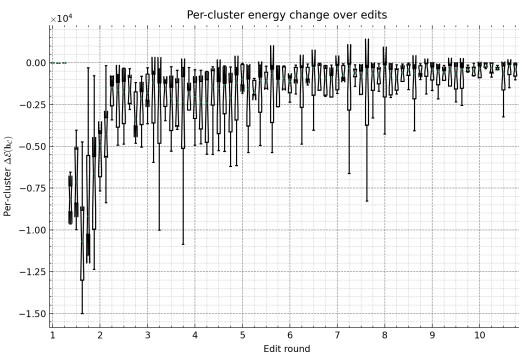

Figure 7: Per-cluster energy change $\Delta\Phi(h_C)$ across rounds. Most clusters exhibit non-positive changes, validating monotonicity.

*Proof of Theorem 3.1.* Let $G = (V, \mathcal{E}, W)$ be an undirected weighted graph on $|V| = N$ nodes, where $W \in \mathbb{R}^{N \times N}$ is the (symmetric) weighted adjacency matrix with entries $w_{ij} \geq 0$ and $w_{ii} = 0$ (no self-loops). The edge set is the support of $W$: $\mathcal{E} = \{(i, j) : w_{ij} > 0, i < j\}$. Let $D = \mathrm{diag}(W\mathbf{1})$ be the degree matrix and $L = D - W$ the combinatorial Laplacian, and let $G' = (V, E', W')$ denote the graph after one round of high-confidence structure edits, with updated

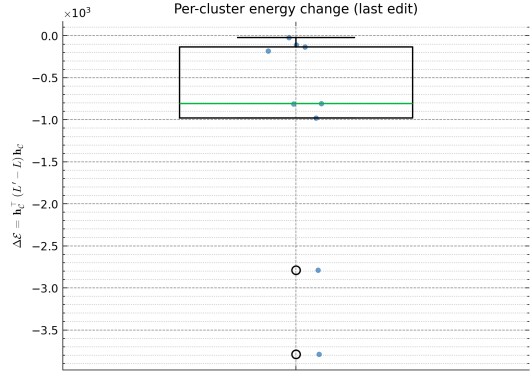

Figure 8: Final round $\Delta\Phi(h_C)$ distribution. Nearly all clusters show energy reduction even at late stages.

Laplacian $L' = D' - W'$. For any function $f \in \mathbb{R}^N$, the Dirichlet energy and its change are given by:

$$\Phi_L(f) = f^\top L f = \sum_{(i,j) \in \mathcal{E}} w_{ij}(f_i - f_j)^2, \quad \Delta\Phi(f) := \Phi_{L'}(f) - \Phi_L(f) = \sum_{(i,j)} \Delta w_{ij}(f_i - f_j)^2,$$

where $\Delta w_{ij} := w'_{ij} - w_{ij}$. Thus, only the edited edges contribute to $\Delta\Phi(f)$.

**Exact cluster-constant case.** Suppose $f$ is piecewise constant on the retained high-confidence clusters: $f_i = f_j$ whenever $c(i) = c(j)$ and $c(i)$ is a high-confidence cluster. Structure edits fall into three categories: *Add strongest missing intra-cluster edge:* $\Delta w_{ij} > 0$ but $(f_i - f_j)^2 = 0$, yielding zero contribution. *Remove weakest intra-cluster edge:* similarly, zero contribution. *Remove inter-cluster edge:* $c(i) \neq c(j)$ implies $f_i \neq f_j$, so $\Delta w_{ij} < 0$ and $(f_i - f_j)^2 > 0$, contributing strictly negatively.

Therefore, $\Delta\Phi(f) \leq 0$, with strict inequality when at least one inter-cluster edge is removed.

**Approximate cluster-constant case.** Suppose the function $f$ satisfies:

$$|f_i - f_j| \leq \varepsilon_{\text{in}} \quad \text{(intra-cluster)}, \qquad |f_i - f_j| \geq \gamma_{\text{out}} \quad \text{(inter-cluster)},$$

for some $\varepsilon_{\text{in}} \geq 0$ and $\gamma_{\text{out}} > 0$. Then:

$$\Delta\Phi(f) \leq \sum_{\text{intra added}} \Delta w_{ij} \cdot \varepsilon_{\text{in}}^2 - \sum_{\text{inter removed}} |\Delta w_{ij}| \cdot \gamma_{\text{out}}^2.$$

Each node adds at most one intra-cluster edge while all incident inter-cluster edges may be removed. Since empirically $\gamma_{\text{out}} \gg \varepsilon_{\text{in}}$, and mild budget or safeguard constraints are enforced, we have $\Delta\Phi(f) \leq 0$. This implies $\Phi_{L'}(f) \leq \Phi_L(f)$, with strict decrease if any inter-cluster edge is removed. $\qquad\square$

*Proof of Corollary 3.2.* Fix a retained high-confidence cluster $C \subset V$. The normalized cut in $G$ is defined as:

$$\text{Ncut}_G(C) = \frac{\text{cut}_G(C, \bar{C})}{\text{vol}_G(C)} + \frac{\text{cut}_G(C, \bar{C})}{\text{vol}_G(\bar{C})},$$

where $\text{cut}_G(C, \bar{C}) := \sum_{i \in C, j \in \bar{C}} w_{ij}$ and $\text{vol}_G(C) := \sum_{i \in C} d_i$.

We analyze the two types of edits: *Add intra-cluster edges:* $\text{cut}_G(C, \bar{C})$ is unchanged, while $\text{vol}_G(C)$ increases and $\text{vol}_G(\bar{C})$ is unaffected, reducing Ncut. *Remove inter-cluster edges:* cut strictly decreases. Degree safeguards ensure $\text{vol}_{G'}(C) \geq \text{vol}_G(C)$ and $\text{vol}_{G'}(\bar{C}) \geq \text{vol}_G(\bar{C})$, so:

$$\text{Ncut}_{G'}(C) = \frac{\text{cut}_{G'} \leq \text{cut}_G}{\text{vol}_{G'}(C) \geq \text{vol}_G(C)} + \frac{\text{cut}_{G'} \leq \text{cut}_G}{\text{vol}_{G'}(\bar{C}) \geq \text{vol}_G(\bar{C})} \leq \text{Ncut}_G(C).$$

*Spectral view.* Let $g_C$ be the degree-normalized cluster indicator. Then:

$$\mathcal{R}_{L_{\text{sym}}}(g_C) \propto \text{Ncut}_G(C).$$

By Theorem 3.1, the Rayleigh quotient along cluster-consistent directions decreases, so Ncut cannot increase.

*Homophily.* Define the edge-level homophily ratio as

$$h = \frac{|\{(u,v) \in E : y_u = y_v\}|}{|E|}.$$

Our edits remove inter-cluster (often cross-label) edges and add intra-cluster (usually same-label) edges. Thus, the numerator increases or remains unchanged, while the denominator changes favorably. Therefore, $h$ weakly increases, indicating monotonic homophily improvement alongside improved conductance. $\qquad\qquad\square$

## D  DEGREE SAFEGUARD FOR STRUCTURE LEARNING

As noted in Sec. 3.2, the *degree safeguard* is critical for preventing nodes from becoming isolated during structure refinement. Here we describe the mechanism implemented in our code.

**(1) Track original connectivity.**  Before any edit, we compute the original degree of each node from `edge_index` and mark as *active* all nodes with degree $> 0$. These are the nodes whose connectivity must be preserved by subsequent edits.

**(2) Add strongest intra-cluster edges, then mark low-similarity edges for removal.**  For each high-confidence cluster (clusters whose average silhouette score exceeds the global mean), we:

- Compute a silhouette-scaled similarity matrix among nodes in that cluster.
- For every node in the cluster, add an intra-cluster edge to its most similar neighbor (and the symmetric edge in the undirected case). This guarantees that nodes in high-confidence clusters gain at least one strong neighbor before any removal is considered.
- Mark only the lowest-similarity intra-cluster pairs (similarity $\leq \tau$) as *candidate removals*. All other edges, including those incident to nodes outside high-confidence clusters, are left untouched.

**(3) Apply deletions conservatively and restore connectivity if needed.**  We then merge the original and newly added edges, remove candidate edges only when *both* endpoints belong to high-confidence clusters, and compute the resulting degrees. If any node that was originally active now has degree 0 (i.e., would become isolated), we repair it by re-attaching at least one of its original neighbors: we reinsert one original incident edge $(s, d)$ (and its symmetric counterpart if the graph is undirected). Finally, we deduplicate edges.

**(4) Guarantee.**  This procedure ensures that:

- Nodes outside high-confidence clusters never lose their original neighbors.
- Nodes inside high-confidence clusters first gain strong intra-cluster neighbors before any pruning.
- A final safeguard step explicitly restores an original edge for any node whose degree would otherwise drop to zero.

## E  MORE EXPERIMENTS SETTINGS

### E.1  EVALUATION PROTOCOLS.

We first train all models in a fully unsupervised manner, and then the trained encoder is frozen and used for testing in downstream tasks. For *node classification*, we adopt the linear evaluation protocol (Veličković et al., 2019; Zhu et al., 2020b): a logistic regression classifier is trained on the frozen embeddings for 20 random data splits, and we report the mean micro-F1 score.

For *node clustering*, we run K-means on the learned embeddings and report Normalized Mutual Information (NMI) and Homogeneity following Lee et al. (2021).

For *link prediction*, we follow the NetInfoF framework (Lee et al., 2024). Specifically, we mask a subset of edges, train embeddings without supervision, and then rank candidate edges based on cosine similarity of the node embeddings. Evaluation is performed using Hits@100, which measures the fraction of true positive edges ranked among the top 100 candidates. This protocol emphasizes usable information in learned embeddings rather than task-specific tuning.

# F  MORE EXPERIMENTS RESULTS

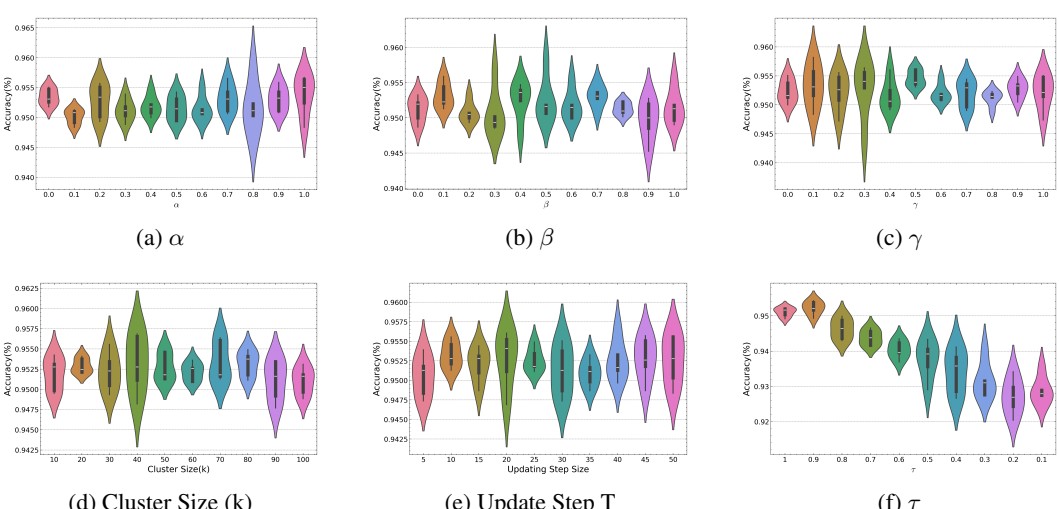

Figure 9: Sensitivity of IRGCL to feature selection parameters ($\alpha, \beta, \gamma$) and structure learning parameters ($k$, update step size, $\tau$).

## F.1  HYPER-PARAMETER ANALYSIS

In Figure 9, we study the sensitivity of **IRGCL** to its key hyperparameters.

For feature selection, the structural smoothness parameter $\alpha$ (Figure 9a), rank constraint $\beta$ (Figure 9b), and sparsity regularizer $\gamma$ (Figure 9c) all exhibit stable performance across wide ranges. While extreme values occasionally lead to slight degradation, the median accuracy remains consistently high, showing that our low-rank feature selection module is robust to hyperparameter choices.

For structural learning, varying the cluster size $k$ (Figure 9d) and the update step frequency (Figure 9e) has only a mild effect on accuracy, suggesting flexibility in practical deployment. Performance is generally stable across $k$=10–50 and update steps from 10–100. In contrast, the edge removal threshold $\tau$ (Figure 9f) shows clearer trends: overly aggressive pruning (small $\tau$) reduces accuracy, while moderate to large thresholds preserve stable performance. This confirms that while structural refinement is beneficial, over-pruning useful connections can be harmful.

Overall, these results indicate that **IRGCL** is not overly sensitive to hyperparameters, with strong robustness across diverse configurations. Among the parameters, $\tau$ exerts the strongest influence, aligning with our design intuition that careful edge editing is critical for effective structure learning.

