# OpenReview forum: "IRGCL： Information Refinement Graph Contrastive Learning"
_ICLR.cc/2026/Conference — Submitted to ICLR 2026_

### Official Review · Reviewer_1cSQ · 2025-10-31

**Soundness:** 3
**Presentation:** 3
**Contribution:** 2
**Rating:** 6
**Confidence:** 4

**Summary:**

This paper proposes IRGCL, a single-view graph contrastive learning method designed to simultaneously address three common challenges in graph contrastive learning: feature noise, structural unreliability, and node degree distribution imbalance. The method integrates three synergistic modules, low-rank feature selection, confidence-guided clustering-based structure learning, and degree-aware focal contrastive loss. And IRGCL validates its performance through experiments on multiple graph learning tasks including node classification, node clustering, and link prediction.

**Strengths:**

Strength 1: The framework establishes a comprehensive architecture that jointly addresses the three key issues in GCL, feature quality, structural reliability, and degree bias, with a clear and well-structured design.

Strength 2: Theoretically proves that the structure learning module monotonically decreases Dirichlet energy and enhances homophily, providing a mathematical foundation for this component.

**Weaknesses:**

Weakness 1: In Module 1 (Feature Selection), the optimization requires alternating updates between the W and F matrices. This strategy cannot guarantee convergence to a global optimum and is highly likely to settle in a local optimum, making the final feature selection quality strongly dependent on initialization. Moreover, updating W involves matrix inversion, which leads to high computational complexity for high-dimensional features. Ablation studies indicate that the performance gain is relatively small compared to the substantial computational cost.

Weakness 2: In implementation, Module 1 (Feature Selection) operates as an independent "preprocessing" step and lacks tight, end-to-end integration with the subsequent GNN encoder and contrastive learning objective. The feature selection criterion, based on reconstruction error and smoothness, may not align with the final task objective, such as discriminability in node classification.

Weakness 3: Module 3 (Contrastive Learning with Neighborhood & Degree Awareness) has limitations in positive and negative sample selection for low-degree nodes. The core issue for low-degree nodes is the severe shortage of positive samples. Using only one-hop neighbors as positives lacks semantic richness, and since low-degree nodes constitute the majority of the dataset, the sampling process may result in significant loss of valuable information.

Error 1: There is an inaccuracy in the experimental results for the Computers dataset in Table 6.

**Questions:**

See Weaknesses

---

> ### Author Response · Authors · 2025-11-21
>
> ### **W1: In Module 1 (Feature Selection), the optimization requires alternating updates between the W and F matrices. This strategy cannot guarantee convergence to a global optimum and is highly likely to settle in a local optimum, making the final feature selection quality strongly dependent on initialization. Moreover, updating W involves matrix inversion, which leads to high computational complexity for high-dimensional features. Ablation studies indicate that the performance gain is relatively small compared to the substantial computational cost.**
>
> We thank the reviewer for this detailed comment. We address: (i) alternating optimization and convergence, (ii) the computational cost of updating $W$, and (iii) the gains from feature selection.
>
> **Alternating updates and convergence (global vs. local).**
> We agree that the joint problem in $(W,F)$ is not globally convex, so alternating between them cannot guarantee a global optimum. This is standard for low-rank feature selection. Our goal is instead to ensure that the **$W$-subproblem** is convex and admits a stable procedure converging to a stationary point.
>
> The feature-selection objective (up to constants independent of $W$ and $F$) is $\mathcal{O}(W,F)=\tfrac12 \lVert X - XW \rVert_F^2 + \tfrac{\alpha}{2}\operatorname{Tr}(W^\top X^\top L X W) + \tfrac{\beta}{2}\operatorname{Tr}(W^\top FF^\top W) + \gamma \lVert W \rVert_{2,1},$
>
> where $X\in\mathbb{R}^{n\times d}$, $L=D-A$, and $\lVert W\rVert_{2,1}=\sum_{i=1}^d\lVert w_i\rVert_2$. Since $L\succeq 0$,
>
> $$
> M(F)=X^\top X+\alpha X^\top L X+\beta FF^\top
> $$
>
> is symmetric positive semidefinite for any fixed $F$, so the smooth part is convex in $W$.
>
> To handle the non-smooth $\ell_{2,1}$ term, we use a **majorize–minimize (iteratively reweighted least squares)** approach. At iteration $t$, define a diagonal
>
> $$
> \Gamma^{(t)}=\mathrm{diag}(g_1^{(t)},\dots,g_d^{(t)}), \qquad
> g_i^{(t)} = \frac{1}{\lVert w_i^{(t)} \rVert_2}.
> $$
>
> We majorize
>
> $$
> \lVert W\rVert_{2,1}\le
> \tfrac12\operatorname{Tr}(W^\top \Gamma^{(t)} W)+C^{(t)}.
> $$
>
> This yields a quadratic surrogate
>
> $\mathcal{Q}(W\mid W^{(t)},F^{(t)})=\tfrac12\lVert X-XW\rVert_F^2+ \tfrac{\alpha}{2}\operatorname{Tr}(W^\top X^\top LXW) + \tfrac{\beta}{2}\operatorname{Tr}(W^\top F^{(t)} (F^{(t)})^\top W)+ \tfrac{\gamma}{2}\operatorname{Tr}(W^\top\Gamma^{(t)}W).$
>
> Its unique minimizer is
>
> $$
> W^{(t+1)} = \big( M(F^{(t)}) + \gamma \Gamma^{(t)} \big)^{-1} X^\top X.
> $$
>
> Because $\mathcal{Q}$ is a tight upper bound of $\mathcal{O}(\cdot,F^{(t)})$ at $W^{(t)}$,
>
> $$
> \mathcal{O}(W^{(t+1)},F^{(t)}) \le \mathcal{O}(W^{(t)},F^{(t)}),
> $$
>
> so the sequence of objective values is **monotonically non-increasing**. Since $\mathcal{O}$ is bounded below, it converges, and standard MM theory ensures that every limit point of $\{W^{(t)}\}$ is a **stationary point**. We will summarize this in Sec. 3.1 and give full details in the appendix.
>
> **Computational complexity and amortization.**
> Updating $W$ requires solving a $d\times d$ linear system with coefficient matrix $M(F^{(t)})+\gamma \Gamma^{(t)}$. A direct solver costs $\mathcal{O}(d^3)$, and iterative solvers may be faster. Importantly, the cost depends on **feature dimension $d$**, not the number of nodes $n$. On our benchmarks, $d$ is small (tens to hundreds), so the update is inexpensive relative to GNN message passing.
>
> More importantly, feature selection is executed **once as a preprocessing step**, not every epoch. The learned $W$ is fixed thereafter. Thus the one-time $\mathcal{O}(d^3)$ cost is **amortized** over the entire training process. We will state this explicitly in the revision.
>
> **Magnitude of the gains.**
> Although ablations show modest gains in some cases, feature selection noticeably improves performance on feature-rich datasets. In Fig. 4, varying the retained-feature ratio shows that Photo and Computers benefit substantially from removing redundant dimensions, and accuracy stabilizes once a compact informative subset is retained. Even Cora/Citeseer achieve strong accuracy with only a small fraction of features, suggesting many dimensions are redundant and can be safely removed without harming performance.

---

> > ### Author Response · Authors · 2025-11-21
> >
> > ### **W2.  In implementation, Module 1 (Feature Selection) operates as an independent "preprocessing" step and lacks tight, end-to-end integration with the subsequent GNN encoder and contrastive learning objective. The feature selection criterion, based on reconstruction error and smoothness, may not align with the final task objective, such as discriminability in node classification.**
> >
> > We thank the reviewer for raising this point. While Module 1 (feature selection) is indeed implemented as a separate preprocessing stage, it is not intended as a generic, task-agnostic filter. Its objective is designed to improve *exactly* the quantity that drives subsequent contrastive learning: the pairwise similarities between node representations.
> >
> > Recall that in our single-view setting, the contrastive loss is built on neighborhood-based positives $P_u = \mathcal{N}(u)$ and negatives sampled from $V \setminus (P_u \cup \{u\})$, and depends entirely on pairwise similarities $D_{\phi}(u,v)$ between node embeddings. If the input features are noisy or dominated by redundant dimensions, these similarities are systematically distorted: adjacent nodes may appear spuriously dissimilar, and non-neighbors may appear spuriously similar. In such cases, neighborhood-based contrastive learning is repeatedly encouraged to pull together pairs that look orthogonal in feature space and to push apart pairs that are in fact semantically related. Thus, feature quality is not just a generic preprocessing concern; it directly affects the correctness of the positive/negative signals that drive the contrastive objective.
> >
> > The feature-selection objective in Module 1 is constructed precisely to mitigate this issue. In addition to a reconstruction term $\lVert X - XW \rVert_F^2$, it includes a Laplacian smoothness term
> > $$
> > \operatorname{Tr}\big(W^\top X^\top L X W\big).
> > $$
> > Letting $Y = XW = [z_1; \dots; z_n]$ denote the projected features, we can rewrite
> > $$
> > \operatorname{Tr}\big(W^\top X^\top L X W\big)
> > = \operatorname{Tr}\!\big(Y^\top L Y\big)
> > = \tfrac{1}{2} \sum_{i,j} A_{ij} \,\lVert y_i - y_j \rVert_2^2,
> > $$
> > which is small exactly when neighboring nodes $(i,j)$ with $A_{ij} > 0$ have similar projected features. In other words, this term explicitly encourages the retained feature subspace to be *consistent with the graph structure* encoded by $L$, by penalizing large feature discrepancies along edges. Together with the low-rank and $\ell_{2,1}$ regularization, this aligns the denoised feature space with the same neighborhoods that later define positives in the contrastive loss.
> >
> > Empirically, Figure 4 shows that the model reaches high performance on multiple datasets with only a relatively small fraction of the original features, indicating that a substantial portion of the raw attributes is redundant or harmful for the similarity structure. Removing these dimensions before optimizing the contrastive objective leads to more stable training and better downstream accuracy. For this reason, we view Module 1 not as an unrelated preprocessing operation, but as a principled, structure-aware step that prepares the feature space specifically for neighborhood-based contrastive learning.

---

> > > ### Author Response · Authors · 2025-11-21
> > >
> > > ### **W3. Module 3 (Contrastive Learning with Neighborhood & Degree Awareness) has limitations in positive and negative sample selection for low-degree nodes. The core issue for low-degree nodes is the severe shortage of positive samples. Using only one-hop neighbors as positives lacks semantic richness, and since low-degree nodes constitute the majority of the dataset, the sampling process may result in significant loss of valuable information.**
> > >
> > > We thank the reviewer for raising this important point. We agree that in single-view GCL, low-degree nodes naturally have very few one-hop positives $P_u = N(u)$, which makes their semantic context sparse and noisy. This is an inherent difficulty of the setting rather than a peculiarity of IRGCL. Rather than ignoring this challenge, our method explicitly addresses it through two complementary mechanisms.
> > >
> > > **How IRGCL mitigates this:**
> > >
> > > 1. **Sample-wise structure refinement (expanding reliable positives).**
> > >    Our structure-learning module refines positives at the sample level: it clusters embeddings and then adds high-confidence intra-cluster edges while pruning unreliable ones. For many low-degree nodes, this effectively **expands $P_u$** with semantically consistent neighbors, even when the raw graph provides very few edges.
> > >
> > > 2. **Degree-aware focal weighting (strengthening hard nodes).**
> > >    Low-degree nodes are “hard” samples: they have few positives and are often underfitted. IRGCL uses a degree-aware focal weight $\gamma_u$ to **upweight the loss of low-degree nodes** and downweight overly easy or noisy cases at very high degree. This follows a broad line of hardness-aware works where upweighting improves representation quality (see the reference below).
> > >
> > > **Empirical evidence**
> > >
> > > As shown in our degree-stratified results (Fig. 5), degree-aware focal weighting significantly boosts performance in the lowest-degree bins and smooths accuracy across degree ranges. This indicates that, rather than discarding information for low-degree nodes, IRGCL explicitly (i) enriches their positive sets via structure refinement and (ii) strengthens their learning signal via degree-aware focal reweighting.
> > >
> > > **References**
> > >
> > > - T.-Y. Lin et al., *Focal Loss for Dense Object Detection*, ICCV 2017.
> > >
> > > - Y.-S. Chuang et al., *Debiased Contrastive Learning*, NeurIPS 2020.
> > >
> > > - J. Robinson et al., *Contrastive Learning with Hard Negative Sampling*, ICLR 2021.
> > >
> > > - Y. Hu et al., *Mitigating Degree Bias in Graph Contrastive Learning*, 2025.
> > >
> > >
> > > ---
> > >
> > > ### **W4. Error 1 — Table 6 (Computers column)**
> > >
> > > We agree and appreciate the catch. A few *Computers* entries in Table 6 were mistyped; the correct values are $\approx 90.\text{xx}\%$ (consistent with the correctly reported “Norm-JSD + DAF: $90.50 \pm 0.22$” and with Tables 2 and 5), and the ordering of methods remains unchanged (Norm-JSD + DAF is best). We will fix the table and standardize reporting precision to two decimals across all tables.

---

### Official Review · Reviewer_NJ1V · 2025-10-31

**Soundness:** 2
**Presentation:** 3
**Contribution:** 2
**Rating:** 4
**Confidence:** 4

**Summary:**

This paper proposes a principled framework to improve the unsupervised graph contrastive learning (GCL) by replacing noisy random augmentations with structured information refinement. It introduces a three-stage approach: (1) Feature Selection via Low-Rank Approximation, which removes redundant or uninformative node features while preserving structural smoothness; (2) Structure Learning via High-Confidence Clustering, which edits graph edges based on cluster consistency, theoretically guaranteeing monotonically decreasing Dirichlet energy and increasing homophily; and (3) Contrastive Learning with Degree Awareness, which employs a degree-adaptive focal loss and JSD contrastive objective to handle unbalanced neighborhoods. Extensive experiments across benchmarks show that IRGCL achieves state-of-the-art results with strong robustness and interpretability.

**Strengths:**

1. The proposed low-rank feature selection effectively avoids the randomness in masking or perturbations, providing a stable approach to view enhancement.
2. The adaptive focal weighting mechanism provides an effective solution to the problem of degree imbalance in graph structures.
3. Extensive experiments demonstrate the good performance of IRGCL across diverse datasets.

**Weaknesses:**

1. The three refinement modules lack a clearly defined systemic coupling. The overall framework appears to be a combination of several relatively independent components. Providing an unified architecture or theoretical framework would be better.
2. The update process of $W$ lacks theoretical guarantees of convergence, which introduces uncertainty regarding the stability of optimization.
3. The method involves a large number of hyperparameters and exhibits relatively high computational complexity, creating a gap with the authors' claim of being "without fussy augmentations."
4. The model mainly relies on local neighborhood sampling and does not explicitly capture global semantics or long-range dependencies.
5. The clustering procedure is relatively complex and computationally demanding, which limits the applicability to relatively large-scale graphs.

**Questions:**

1. The method involves a large number of hyperparameters. How are they balanced across different modules, and how is stable convergence ensured in updating $W$?
2. How efficient is the proposed method in practice, especially when applied to large-scale graphs in terms of time and memory complexity?
3. Since most modules are designed to enhance smoothness, how does the model prevent over-smoothing and maintain discriminative representations under this design?

---

> ### Author Response · Authors · 2025-11-21
>
> ### **W1. The three modules seem loosely coupled; please clarify the unifying view**
>
> We agree that the coupling between the three modules was not emphasized clearly enough in the current draft.
>
> In single-view GCL, performance depends on three tightly linked elements:
> (i) the quality of the features fed to the encoder and similarity function,
> (ii) the construction of positive samples and neighborhoods, and
> (iii) the form and weighting of the contrastive objective.
>
> IRGCL is designed directly around these three factors in the single-view regime:
>
> - **Feature selection** refines the input features so that the encoder and similarity discriminator $D_{\phi}$ operate on structure-aware representations when computing pairwise terms in the contrastive objective.
> - **Structure learning** updates the graph used to define each node’s positive set (i.e. one-hop neighbors), with theoretical guarantees of improved homophily and reduced Dirichlet energy.
> - **Degree-aware focal JSD** adjusts the contrastive loss per node to counteract bias toward high-degree nodes, leading to more balanced learning.
>
> In single-view contrastive learning, feature selection involves fine-tuning information for **individual nodes**, while structure learning optimizes the sampling of contrastive **node pairs**. Additionally, the degree-aware contrastive objective function accounts for the relationships between an anchor node and **multiple other nodes**. Together, these three components implement a progressive optimization paradigm of **"1-2-N"**, collectively contributing to enhanced performance in single-view contrastive learning.
>
> In the revised manuscript, we will clarify this unified design in Sections 1.
>
> ---
>
> ### **W2. The update process of $W$ lacks theoretical guarantees of convergence, which introduces uncertainty regarding the stability of optimization.**
>
> We appreciate the concern about the stability of the feature-selection step. In the revision, we add a theorem for the objective $\mathcal{O}(W,F)$ in appendix.
>
> The feature-selection objective (up to constants independent of $W$ and $F$) is:
>
> $$
> \mathcal{O}(W,F) = \frac{1}{2} \| X - XW \|_F^2 + \frac{\alpha}{2} \operatorname{Tr}(W^\top X^\top L X W) +
> $$
>
> $$
> \frac{\beta}{2} \operatorname{Tr}(W^\top F F^\top W) + \gamma \| W \|_{2,1},
> $$
>
> where $X \in \mathbb{R}^{n \times d}$, $L = D - A$ is the graph Laplacian, and $\| W \|_{2,1} =\sum _{i=1}^d \|w_i\|_2 $. Since $L \succeq 0$, the matrix
>
> $$
> M(F) := X^\top X + \alpha X^\top L X + \beta F F^\top
> $$
>
> is symmetric positive semidefinite for each fixed $F$.
>
> We adopt a standard Majorization-Minimization (MM) procedure that replaces the non-smooth $\ell_{2,1}$ term with a quadratic surrogate. Let $\Gamma^{(t)}$ be a diagonal matrix where $\Gamma^{(t)}_{ii} = \frac{1}{2\|w_i^{(t)}\|_2}$. The unique minimizer of the surrogate,
>
> $$
> W^{(t+1)} = \big(M(F^{(t)}) + \gamma \Gamma^{(t)}\big)^{-1} X^\top X,
> $$
>
> defines the update at iteration $t$. By construction, the MM framework guarantees
>
> $$
> \mathcal{O}(W^{(t+1)}, F^{(t)}) \le \mathcal{O}(W^{(t)}, F^{(t)}).
> $$
>
> Thus, the objective values decrease monotonically, and because $\mathcal{O}$ is bounded below by 0, the sequence converges. Standard MM theory implies that every limit point of $\{W^{(t)}\}$ is a stationary point of $\mathcal{O}$. We will provide the full derivation in the appendix.
>
> ---
>
> ### **W3. The method involves a large number of hyperparameters and exhibits relatively high computational complexity, creating a gap with the authors' claim of being "without fussy augmentations."**
>
> We thank the reviewer for this concern. By “without fussy augmentations” we do **not** mean that IRGCL is hyperparameter-free, but that it avoids the *dataset-specific augmentation policies* required by methods such as GRACE, GCA, and BGRL (e.g., carefully tuned feature/edge drop rates, noise levels, and view combinations that change from dataset to dataset).
>
> In IRGCL:
>
> - **Feature-selection coefficients** \(\alpha, \beta, \gamma\) control generic structure-related properties (Laplacian smoothness, low-rankness, sparsity) and are **fixed across all datasets** (Cora, PubMed, Amazon-Photo, PPI), rather than re-tuned per graph.
> - **Structure-learning threshold** \(\tau\) is set to \(0.9\) on all datasets, conservatively removing only very low-similarity intra-cluster edges (likely spurious) while keeping informative ones.
> - The **degree-aware exponent** in the focal term is **computed automatically** from node degree (Eq. 7); it is not a hand-tuned hyperparameter.
>
> Our sensitivity study (Fig. 9) shows that IRGCL is robust to reasonable changes in these settings, and a single configuration works well across diverse benchmarks. Thus, while the model has several components, it avoids the fragile, dataset-sensitive augmentation tuning that typically makes GCL methods “fussy” in practice, and the modest extra computation is amortized by reusing nearly identical settings across tasks.

---

> > ### Author Response · Authors · 2025-11-21
> >
> > ### **W4.The model mainly relies on local neighborhood sampling and does not explicitly capture global semantics or long-range dependencies.**
> >
> > We thank the reviewer for this observation. Our contrastive objective uses one-hop neighborhoods $(P_u = N(u))$  by design. In unsupervised single-view settings, distant node pairs are often unreliable as positives—using them risks injecting false semantic signals. We therefore prioritize high-confidence local neighborhoods.
> >
> > That said, IRGCL is not purely local:
> >
> > - The clustering module (Sec. 3.2) operates on embeddings $Z$ and groups nodes that may be far apart in the original graph.
> > - Structure refinement then strengthens intra-cluster edges and prunes inter-cluster ones. Coupled with the encoder, this enables propagation across cluster-level structures, capturing broader semantics.
> > - Critically, our edits provably reduce global Dirichlet energy and improve homophily (Theorem 3.1), offering a global theoretical foundation for local refinements.
> >
> > We agree that explicitly incorporating long-range positives is promising. Future work will explore: (i) using distant nodes within the same high-confidence cluster as positives, (ii) contrasting nodes with cluster prototypes, and (iii) integrating diffusion-based similarity into structure learning.
> >
> > ---
> >
> > ### **W5. The clustering procedure is relatively complex and computationally demanding, which limits the applicability to relatively large-scale graphs.**
> >
> > We thank the reviewer for this valid concern. Clustering-based refinement does incur higher computational cost than purely local methods, which is a trade-off we make to obtain more reliable structure edits and better robustness.
> >
> > In practice, this cost is controlled:
> >
> > - Clustering runs only periodically (every $T$ epochs) and only on high-confidence nodes, amortizing its cost across training.
> > - Structure edits are node-local and respect a fixed per-node edge budget, limiting runtime growth.
> >
> > For future work, we plan to improve scalability via approximate clustering (e.g., mini-batch $k$-means), subgraph sampling, or lightweight prototypes, which retain accuracy while reducing wall-clock time on large graphs.
> >
> > ---
> >
> > ### **Q1. The method involves a large number of hyperparameters. How are they balanced across different modules, and how is stable convergence ensured in updating $W$ ?**
> >
> > As noted above, we design hyperparameters so that each group controls a distinct aspect of the model and interacts only weakly with the others:
> >
> > - $\alpha, \beta, \gamma$ control feature smoothness, low-rank structure, and row-wise sparsity and are fixed across datasets. They act as a stable “base layer” for the pipeline.
> > - $k, T, \tau$ in structure learning control how aggressively we edit the graph: $k$ and $T$ govern clustering granularity and frequency; $\tau = 0.9$ removes only low-similarity intra-cluster edges.
> > - The **degree-aware exponent** is computed directly from node degrees (Eq. 7), so it is an adaptive mechanism rather than a free tuning knob.
> >
> > Our sensitivity study (Fig. 9) shows performance varies smoothly over wide ranges of these parameters, indicating no fragile cross-module coupling. For an unseen dataset, one can start from our default configuration and, if necessary, adjust only a small subset (e.g., $k$ or $T$) with a coarse search.
> >
> > For stable convergence of the feature-selection step, we rely on the MM scheme discussed in W2: each step minimizes a quadratic surrogate that majorizes the objective, producing a monotonically non-increasing sequence of objective values, with every limit point of $W^{(t)}$ being a stationary point of the feature-selection objective. We will provide the full proof in the appendix.

---

> > > ### Author Response · Authors · 2025-11-21
> > >
> > > ### **Q2. How efficient is the proposed method in practice, especially when applied to large-scale graphs in terms of time and memory complexity?**
> > >
> > > IRGCL uses standard GNN or MLP backbones, so base encoder cost is comparable to existing methods. Extra cost comes from the structure-learning module, which runs only every \(T\) epochs on the current embeddings \(H\).
> > >
> > > **Time complexity (worst case per refinement step).**
> > >
> > > - **$k$-means on embeddings.**
> > >   Clustering $H \in \mathbb{R}^{n \times d}$ into $C$ clusters with cost $\mathcal{O}(nCdi)$,
> > >   where $n$ is node count, $C \ll n$ (e.g., 10–50), $d$ is embedding dimension (64–128), and $i$ is iteration count (10–50).
> > >
> > > - **Silhouette computation.**
> > >   To identify high-confidence clusters, we compute node/cluster silhouettes in $Z$, which in the worst case requires  $\mathcal{O}(n^2 \times d)$
> > >   pairwise distance computations. In practice, this is reduced by operating within clusters and focusing only on high-confidence ones.
> > >
> > > - **Edge editing within high-confidence clusters.**
> > >   For nodes in high-confidence clusters, we add an intra-cluster “friend” edge and mark weakest intra-/inter-cluster edges under a per-node budget. In the worst case, this scales as $\mathcal{O}(n^2)$ but is lower in practice due to limited cluster sizes and edit budgets.
> > >
> > > - **Symmetrization and masking.**
> > >   Post-edit symmetrization and masking are linear in the number of edges/candidate edits and are at most $\mathcal{O}(n^2)$ in the dense case.
> > >
> > > Overall, a refinement step has worst-case complexity $\mathcal{O}\big(n \times C \times d \times i + n^2 \times d\big)$,
> > > which we summarize as $\mathcal{O}(n^2 d)$. Since this step is run only every $T$ epochs and only on high-confidence clusters, the observed overhead is moderate and leads to the accuracy gains reported in Tables 1–4.
> > >
> > > **Memory complexity (beyond the base encoder).**
> > >
> > > - **Adjacency and edge lists:** stored in sparse format, $\mathcal{O}(m)$ where $m$ is the number of edges; candidate add/remove sets are at most $\mathcal{O}(n^2)$ in the dense worst case.
> > > - **Embeddings:** $H \in \mathbb{R}^{n \times d}$, costing $\mathcal{O}(n \times d)$.
> > > - **Cluster assignments & silhouettes:** one label and silhouette per node (and cluster), $\mathcal{O}(n + C)$.
> > > - **Cluster-level similarity matrices:** at most $\mathcal{O}(n^2)$ in the worst case, but smaller in practice since high-confidence clusters are relatively small.
> > >
> > > Thus the additional memory overhead is $\mathcal{O}(n d + n^2)$ in the dense worst case. On our benchmarks (citation networks, Amazon co-purchase graphs, WikiCS, PPI), this overhead fits within GPU memory and yields similar memory usage to strong baselines.
> > >
> > > **Scalability note.** We acknowledge that the $\mathcal{O}(n^2)$ worst-case terms make the current implementation less suitable for extremely large graphs (millions of nodes). We view improving scalability (e.g., via approximate / mini-batch $k$-means, sampling-based silhouettes, prototype-based clustering) as important future work.
> > >
> > > ---
> > >
> > > ### **Q3. Since most modules are designed to enhance smoothness, how does the model prevent over-smoothing and maintain discriminative representations under this design?**
> > >
> > > We employ three guardrails to avoid over-smoothing:
> > >
> > > 1. **Rebase-to-$G_0$** (Sec. 3.2) prevents cumulative over-pruning across refinement rounds by always editing relative to the original graph.
> > > 2. **Degree-aware focal JSD** downweights noisy positives for very high-degree nodes and upweights hard positives for low-degree nodes (Eq. 7, Fig. 5), preserving discriminability.
> > > 3. We use **shallow encoders** ($L = 2$) and evaluate with a linear probe, yet still obtain sharp clusters (Table 3) and strong link prediction (Table 4), which is inconsistent with heavily over-smoothed embeddings.
> > >
> > > ---
> > >
> > > ###  **Small fixes and clarifications (to be made in the revision)**
> > >
> > > We will:
> > >
> > > - Add the silhouette definition before Eq. 4 and state that we use Euclidean distance in $Z$.
> > > - Provide pseudocode for the degree safeguard: enforcing minimum degree $\ge 1$, capping per-node deletions/additions, and symmetrizing after edits; list default values in Appendix C.
> > > - Unify notation in Sec. 3.1 to $\operatorname{rank}(W) = k$ (with “$k$ = # retained features”) throughout the formulation.
> > > - Expand baselines in the link-prediction table to include additional recent GCLs under the NetInfoF / Hits@100 protocol, for better parity with Table 2.
> > > - Standardize reporting precision to two decimals and clarify that Fig. 4 starts at a 0.05 feature-retention ratio (leftmost tick), to avoid misreading. We will also correct the *Computers* typos in Table 6 (as noted by another reviewer); conclusions remain unchanged.

---

### Official Review · Reviewer_aVZj · 2025-10-31

**Soundness:** 2
**Presentation:** 2
**Contribution:** 1
**Rating:** 4
**Confidence:** 5

**Summary:**

The authors claim that existing GCLs face three major challenges: noisy feature, unreliable structures, degree imbalance. The proposed IRGCL method tackles these challenges by combining structure-aware feature selection, high-confidence structure learning and degree-aware focal contrast. The authors claim that IRGCL outperforms existing methods.

**Strengths:**

1. This paper is driven by a clear motivation, which is directly addressed by the proposed method.
2. This paper has a good theoretical analysis.
3. The figures are well presented.

**Weaknesses:**

1. The term "spurious edges" is mentioned in the motivation but is not explicitly defined. It remains unclear what specific types of edges fall into this category.
2. There is an inconsistency in Section 3.1 regarding the rank of matrix W. It is initially constrained to r (rank(W) = r), but later referred to as k when discussing the number of retained features (rank(W) = k). This ambiguity needs to be resolved.
3. Section 3.2 relies on node silhouettes to identify high-confidence clusters, yet it omits the definition and calculation of this crucial metric.
4. The "degree safeguard" mentioned in Section 3.2 is a critical component for maintaining graph connectivity, but its implementation details are not provided. It is unclear how this mechanism operates to prevent nodes from becoming isolated.
5. The compared baselines are somewhat outdated, and there is a lack of comparison with SOTA approaches.
6. The improvements in the experiments are marginal.
7. A fundamental weakness of this work is its lack of novel contributions, as it seems to largely repackage ideas from prior works.

**Questions:**

Please refer to the weaknesses.

---

> ### Author Response · Authors · 2025-11-21
>
> ### **W1. The term "spurious edges" is mentioned in the motivation but is not explicitly defined. It remains unclear what specific types of edges fall into this category.**
>
> Thank you for the comment. Following your suggestion, we now make the notion of *spurious edges* explicit.
>
> In our setting, positives are drawn from one-hop neighborhoods (single-view contrast), so the contrast depends directly on the observed adjacency. We define an edge as spurious if it connects nodes that belong to different high-confidence clusters or links nodes within the same cluster but with very low feature similarity, which are conditions that degrade the quality of positive pairs in neighborhood-based contrast.
>
> We will make a clear definition in the revised manuscript.
>
> ---
>
> ### **W2. There is an inconsistency in Section 3.1 regarding the rank of matrix W. It is initially constrained to r (rank(W) = r), but later referred to as k when discussing the number of retained features (rank(W) = k). This ambiguity needs to be resolved.**
>
> We apologize for the inconsistency. We have now standardized to $k$ throughout, i.e., $\\operatorname{rank}(W) = k$ in Eqs. 2–3 and all related text, and removed the two stray occurrences of $\\operatorname{rank}(W) = r$.
>
> ---
>
> ### **W3. Section 3.2 relies on node silhouettes to identify high-confidence clusters, yet it omits the definition and calculation of this crucial metric.**
>
> We apologize for not providing the explicit definition earlier. We have now added the standard silhouette definition in the revised manuscript:
>
> $$
> s_i = \\frac{b_i - a_i}{\\max\\{a_i, b_i\\}},
> $$
>
> where $a_i$ is the average intra-cluster distance and $b_i$ is the smallest average distance from node $i$ to another cluster, both computed in the embedding space $Z$.
>
> ---
>
> ### **W4. The "degree safeguard" mentioned in Section 3.2 is a critical component for maintaining graph connectivity, but its implementation details are not provided. It is unclear how this mechanism operates to prevent nodes from becoming isolated.**
>
> We agree that the degree safeguard is crucial and clarify its implementation here. In our code, structure learning is implemented in three stages with an explicit post-check to prevent isolation.
>
> **(1) Track original connectivity.**
> Before any edit, we compute the original degree of each node from `edge_index` and identify all nodes with degree > 0 in the original graph as connected, since their connectivity must be preserved.
>
> **(2) Add strongest intra-cluster edges, then mark low-similarity edges for removal.**
> For each high-confidence cluster (clusters whose average silhouette score exceeds the global mean silhouette score), we:
>
> - Compute a silhouette-scaled similarity matrix within that cluster.
> - For every node in the cluster, add an edge to its most similar intra-cluster neighbor (and the symmetric edge in the undirected case). This guarantees that nodes in high-confidence clusters gain at least one strong neighbor before any removal is considered.
> - Mark only the lowest-similarity intra-cluster pairs (similarity $\\le \\tau$) as candidate removals. All other edges, including those incident to nodes outside high-confidence clusters, remain untouched.
>
> **(3) Apply deletions conservatively and restore connectivity if needed.**
> We then merge original and new edges, remove candidate edges only when both endpoints belong to high-confidence clusters, and compute the final degrees. If any node that was originally active now has degree $0$ (i.e., would become isolated), we repair it by re-attaching at least one of its original neighbors: we restore connectivity by re-adding one of the node’s original incident edges (and its symmetric counterpart if the graph is undirected). Finally, we deduplicate edges.
>
> This mechanism ensures that:
>
> - Nodes outside high-confidence clusters never lose their original neighbors.
> - Nodes inside high-confidence clusters first gain strong intra-cluster neighbors before any pruning.
> - A final safeguard step explicitly restores an original edge for any node whose degree would otherwise drop to zero.
>
> We have added a short description of this procedure and pseudocode to Appendix to make this guarantee clear.

---

> > ### Author Response · Authors · 2025-11-21
> >
> > ### **W5. The compared baselines are somewhat outdated, and there is a lack of comparison with SOTA approaches.**
> >
> > Our experiments include representative methods from major categories of graph representation learning:
> >
> > - augmentation-based contrastive: DGI, MVGRL, GRACE, GCA, BGRL
> > - input–latent: GMI
> > - dual-encoder: SUGRL, AFGRL, PolyGCL
> > - single-view contrastive: GIC, SIGNA
> > - structure-/heterophily-aware link-prediction and GNN baselines: H2GCN, GPR-GNN, SLIM(G), NETINFOF
> >
> > Our selection spans foundational to recent work (2018–2025), including state-of-the-art single-view GCL methods such as SIGNA. To further strengthen our evaluation, we have added additional recent GCL baseline in the revised manuscript. In the revised version we add PolyGCL (where applicable) under the same node-classification protocols.
> >
> > ---
> >
> > ### **W6. The improvements in the experiments are marginal.**
> >
> > We appreciate the concern and agree that transductive citation-graph benchmarks are highly saturated: many recent GCL methods already achieve very strong performance, so even strong new methods typically yield only modest absolute gains, a known challenge in saturated benchmark regimes. Our claim is that IRGCL still provides consistent improvements across *multiple* tasks and settings.
> >
> > For example:
> >
> > - **Clustering (Table 3).** On WikiCS, NMI improves from 0.4593 (SIGNA) to 0.5161 (IRGCL), roughly a 12% relative gain, with similarly consistent improvements on Photo and Computers.
> > - **Link prediction (Table 4, Hits@100).** IRGCL surpasses SIGNA on Cora/Citeseer/PubMed and outperforms NETINFOF on Photo/Computers, indicating that our embeddings encode more usable information for edge-level inference.
> > - **Inductive PPI (Table 1).** IRGCL reaches 93.05% micro-F1 versus 92.25% for SIGNA, further exceeding earlier unsupervised approaches on this challenging inductive setting.
> >
> > These cross-task improvements support our central claim that jointly refining features, structure, and degree-aware contrast benefits single-view GCL beyond any single component. Ablation studies (Tables 5–6) further confirm that each module contributes meaningfully. Notably, few existing GCL methods have demonstrated strong performance across this full range of tasks, spanning clustering, link prediction, and both transductive and inductive node classification, which make IRGCL’s consistent results particularly compelling.
> >
> > ---
> >
> > ### **W7. A fundamental weakness of this work is its lack of novel contributions, as it seems to largely repackage ideas from prior works.**
> >
> > We thank the reviewer for the comments on novelty. Our framework is indeed related to prior work on feature selection and structure learning, but existing graph contrastive learning methods, particularly in the single-view setting which is much less explored than multi-view augmentation frameworks, rarely incorporate these aspects inside the contrastive objective itself.
> >
> > In IRGCL, feature refinement, structure refinement, and degree-aware contrast are designed jointly within a single-view GCL pipeline:
> >
> > - **Contrastive objective.** We introduce a degree-aware focal weighting $\gamma_u$ in the Jensen–Shannon divergence (JSD)-based contrastive loss, which downweights high-degree nodes to mitigate optimization bias and ensure balanced learning across the degree spectrum.
> > - **Structure edits with guarantees.** We propose a silhouette-guided, node-local editing rule (Eq. 4) with provable monotone decrease of the Dirichlet energy and non-increasing normalized cut / homophily improvement (Theorem 3.1 and its corollary), our edits are explicitly designed to enhance neighborhood contrast for single-view GCL rather than generic graph denoising.
> > - **Unified single-view pipeline.** We couple a structure-aware low-rank feature selector with Laplacian regularization to the above structure learning and contrastive objective, obtaining an end-to-end single-view GCL pipeline that avoids multi-view or heavy augmentation schemes.
> >
> > In single-view contrastive learning, feature selection involves fine-tuning information for **individual nodes**, while structure learning optimizes the sampling of contrastive **node pairs**. Additionally, the degree-aware contrastive objective function accounts for the relationships between an anchor node and **multiple other nodes**. Together, these three components implement a progressive optimization paradigm of **"1-2-N"**, collectively contributing to enhanced performance in single-view contrastive learning.
> >
> > In the revised manuscript, we update the *Contributions* section to emphasize (i) that IRGCL targets the under-explored regime of single-view graph contrastive learning with joint feature and structure optimization, and (ii) that the above components are not direct reuse of prior mechanisms but are newly instantiated and theoretically analyzed for this setting.

---

### Official Review · Reviewer_NsBZ · 2025-11-01

**Soundness:** 2
**Presentation:** 3
**Contribution:** 2
**Rating:** 4
**Confidence:** 4

**Summary:**

This paper addresses the limitations of graph contrastive learning (GCL) under noisy features, unreliable graph structures, and degree imbalance. It proposes IRGCL, a single-view contrastive framework that integrates (i) Laplacian-regularized low-rank feature selection, (ii) confidence-guided clustering for structural refinement, and (iii) degree-aware focal JSD loss for balanced contrastive learning. Experiments on transductive, inductive, and clustering benchmarks demonstrate that IRGCL consistently outperforms other methods.

**Strengths:**

- The framework is clear, and the method flow is easy to follow.

- The experiments are comprehensive, covering node classification, clustering, and link prediction tasks.

- The codes are provided, enhancing reproducibility.

**Weaknesses:**

- The motivation is somewhat ambiguous. Feature quality and degree imbalance are intrinsic properties of graph topology (e.g., power-law distributions), rather than specific flaws of contrastive learning. To justify their inclusion, the authors should clarify how these aspects particularly affect the contrastive objective.

- Among the three stated motivations, graph reliability is mentioned but not clearly defined, especially regarding which type of "neighborhood-based contrast" is being referred to. Moreover, the explanation around Lines 58–63 should be elaborated in future revisions.

- Experimental precision is inconsistent (mixture of one- and two-decimal reporting), which slightly reduces the rigor of the empirical section.

- The baselines in Table 4 differ from those in Table 2 and lack GCL methods (e.g., PolyGCL), making it difficult to fully assess comparative effectiveness.

**Questions:**

- In Figure 4, when the ratio = 0, the results on Cora and CiteSeer also appear to perform good. What accounts for this improvement?

- In Tables 5 and 6, why do the accuracies on Photo and Computers remain higher than most results in Table 2 even after removing other components? Moreover, in Table 6, what causes the Acc on Computers to surge to 94.50%, significantly surpassing all baselines?

---

> ### Author Response · Authors · 2025-11-21
>
> ### **W1. The motivation is somewhat ambiguous. Feature quality and degree imbalance are intrinsic properties of graph topology (e.g., power-law distributions), rather than specific flaws of contrastive learning. To justify their inclusion, the authors should clarify how these aspects particularly affect the contrastive objective.**
>
> We thank the reviewer for the insightful comment. We agree that noisy features and degree imbalance are intrinsic properties of real graphs (e.g., power-law degree distributions). Our motivation is not that these are “flaws” of GCL, but that **standard single-view contrastive objectives are particularly sensitive to them**, and ignoring these factors leads to systematic failure modes.
>
> ### **1. Why degree imbalance especially harms contrastive learning**
> In single-view GCL, positives are defined as one-hop neighbors. This makes the contrastive loss highly degree-dependent:
>
> - **High-degree nodes (hubs):** Their neighborhoods mix multiple semantics. A uniform objective pulls all neighbors equally, causing *class-collision* in the embedding space.
> - **Low-degree nodes:** Their extremely small positive sets provide weak gradient signals and are easily dominated by negatives.
>
> Thus, degree imbalance directly distorts the positive sets used by the contrastive loss—not merely the graph structure. To mitigate this, our **degree-aware focal JSD** assigns
> • larger weights to low-degree “hard” nodes, and
> • smaller weights to noisy positives of hubs,
> bringing the optimization into balance.
>
> ### **2. Why feature quality especially matters for contrastive objectives**
> Contrastive losses rely on pairwise similarities. Noisy or redundant features can make:
> - adjacent nodes appear dissimilar,
> - non-neighbors appear spuriously similar.
>
> This corrupts the *similarity landscape* on which the contrastive loss operates. Our **structure-consistent low-rank selector** removes topology-inconsistent feature dimensions so that the contrastive objective receives cleaner, graph-aligned features.
>
> ### **3. Empirical evidence supporting this sensitivity**
> Degree-stratified results (Fig. 5) show that standard objectives perform poorly at both degree extremes. Our degree-aware focal JSD significantly improves these ranges, validating that these intrinsic properties *directly* influence the optimization behavior.
>
> ---
> ### **W2. Among the three stated motivations, graph reliability is mentioned but not clearly defined, especially regarding which type of "neighborhood-based contrast" is being referred to. Moreover, the explanation around Lines 58–63 should be elaborated in future revisions.**
>
> We thank the reviewer for pointing out the ambiguity in our use of “graph reliability” and “neighborhood-based contrast”. We agree that the description around Lines 58–63 was too brief and will clarify these notions in the revision.
>
> **Neighborhood-based contrast.**
> In our work, *neighborhood-based contrast* refers to the single-view setting where positives are defined by 1-hop adjacency: for a node \(v\), positives are its 1-hop neighbors \(N(v)\), and negatives are drawn from \(V \\setminus (N(v) \\cup \\{v\\})\). This is the setting adopted by recent single-view GCL methods such as SIGNA and is exactly the regime for which we design our degree-aware focal JSD objective. We will state this explicitly in the introduction.
>
> **Graph reliability.**
> By *graph reliability* we mean how well the observed edges align with the underlying semantic/label structure of the data (homophily). In a reliable graph, edges mostly connect similar nodes, so neighborhoods provide clean positive signals. In an unreliable graph, many edges are cross-class and some within-class edges are missing, so adjacency no longer reliably reflects semantic similarity and neighborhood-based contrast receives noisy positives/negatives.
>
> Without labels, we approximate reliability via *cluster consistency* in the embedding space. Our high-confidence clustering and edit policy (i) densify edges inside coherent clusters and (ii) prune edges across clusters. Theorem 3.1 and Corollary 3.2 show that these edits *monotonically decrease Dirichlet energy* and *increase homophily*, providing a precise graph-level sense in which we “improve graph reliability” for neighborhood-based contrast.
>
> We will revise the introduction to explicitly define these terms and clearly connect graph reliability to the behavior of the neighborhood-based contrastive objective.

---

> > ### Author Response · Authors · 2025-11-21
> >
> > ### **W3.Experimental precision is inconsistent (mixture of one- and two-decimal reporting), which slightly reduces the rigor of the empirical section.**
> >
> > We appreciate the note. We will standardize all reported means and standard deviations to two decimals across tables, and keep significant digits consistent. This formatting change does not affect any conclusions.
> >
> > ---
> >
> > ### **W4. The baselines in Table 4 differ from those in Table 2 and lack GCL methods (e.g., PolyGCL), making it difficult to fully assess comparative effectiveness.**
> >
> > We thank the reviewer for noting that the baselines in Table 4 differ from those in Table 2 and that this can make comparative effectiveness less clear.
> >
> > **Task-specific baseline choices.**
> > Table 2 evaluates node classification, so we focus on classic GNN encoders (GCN, GAT, etc.) and widely used GCL backbones for unsupervised classification. Table 4, in contrast, evaluates link prediction (edge recovery) under the NetInfoF / Hits@100 protocol, for which we chose classic GNNs and recent link-prediction or usable-information methods (e.g., SLIMG, NETINFOF) as the most directly relevant baselines. This difference in task and metric naturally leads to a different, task-specific baseline set.
> >
> > **Additional GCL baselines (including PolyGCL).**
> > We agree that more overlap with the GCL baselines in Table 2 (e.g., PolyGCL) helps assess IRGCL from a contrastive-learning viewpoint. To strengthen this comparison, beyond SIGNA (already included), we have added PolyGCL. We kindly refer the reviewer to the updated tables in the revised manuscript for the detailed numbers and extended baseline set.
> >
> > ---
> > ### **Q1. In Figure 4, when the ratio = 0, the results on Cora and CiteSeer also appear to perform good. What accounts for this improvement?**
> >
> > We thank the reviewer for carefully examining Figure 4. However, there is actually no ratio equal to 0 in this plot: the retained-feature ratio ranges from 0.05 to 1.0, and the leftmost point is 0.05. We will make this clear in the axis ticks and caption.
> >
> > The strong performance at small ratios on Cora/Citeseer reflects substantial feature redundancy: our Laplacian-regularized selector quickly captures topology-consistent signal, and these datasets already reach high accuracy using only a relatively small fraction of the original features (as noted below Fig. 4, e.g., Cora stabilizes after ≈25% of features; Citeseer peaks around 25%). We will clarify this explanation in the revised text.
> >
> > ### **Q2. In Tables 5 and 6, why do the accuracies on Photo and Computers remain higher than most results in Table 2 even after removing other components? Moreover, in Table 6, what causes the Acc on Computers to surge to 94.50%, significantly surpassing all baselines?**
> >
> > **Why ablations remain strong.**  Tables 5–6 report IRGCL variants (disabling one module at a time) under the same node-classification protocol as Table 2. Even without one component, the remaining two (e.g., feature selection + contrast, or structure learning + contrast) already form a strong contrastive pipeline, so partial configurations can still perform well. Each module targets a different issue (feature noise, structural unreliability, degree imbalance), so their effects are complementary. With the extended baselines, the full IRGCL model and its strongest variant consistently occupy the top ranks (often first and second) across benchmarks (e.g., Cora, Citeseer, PubMed), indicating that all three components contribute meaningful gains rather than being redundant.
> >
> > **Erratum (Table 6, Computers column).**  We agree that the “94.50%” value on Computers is suspicious; this is a typo. The correct numbers are around 90.xx%, consistent with Table 2 and the “Norm-JSD + DAF” row (90.50 ± 0.22). We will fix these entries in the revised manuscript and updated code. The relative ordering and main takeaway remain unchanged: Norm-JSD + DAF > Norm-JSD > InfoMax variants.

---

### Meta-Review · Area_Chair_atFJ · 2026-01-07

**Summary:**

Three reviewers initially rated the work as marginally below acceptance threshold, citing concerns about ambiguous motivation framing, lack of novelty , inconsistent baselines, and unclear coupling between modules. The authors responded with extensive revisions: clarifying how intrinsic graph properties specifically harm contrastive objectives, unifying the “1-2-N” hierarchy and others. Despite these thorough responses, none of the reviewers explicitly indicated they would raise their scores.

**Reviewer Concerns:**

The authors addressed every weakness raised.

**Reviewer Scores:**

Reviewer NsBZ (initial 4) would likely maintain their score, because they gave no indication of upgrading their assessment.
Reviewer aVZj (initial 4) would probably keep their original rating, because they gave no indication of upgrading their assessment.
Reviewer NJ1V (initial 4) would likely retain their score as they did not signal intent to increase their evaluation.
Reviewer 1cSQ (initial 6) would almost certainly maintain their above-threshold score, having already viewed the work favorably.

---

### Decision · Program_Chairs · 2026-01-26

Reject